# Cellular stress signaling activates type-I IFN response through FOXO3-regulated lamin posttranslational modification

Inah Hwang[1], Hiroki Uchida[1], Ziwei Dai [2], Fei Li[3], Teresa Sanchez [1,4], Jason W. Locasale [2], Lewis C. Cantley [5,6], Hongwu Zheng[1] & Jihye Paik[1,6 ✉]

Neural stem/progenitor cells (NSPCs) persist over the lifespan while encountering constant challenges from age or injury related brain environmental changes like elevated oxidative stress. But how oxidative stress regulates NSPC and its neurogenic differentiation is less clear. Here we report that acutely elevated cellular oxidative stress in NSPCs modulates neurogenic differentiation through induction of Forkhead box protein O3 (FOXO3)-mediated cGAS/STING and type I interferon (IFN-I) responses. We show that oxidative stress activates FOXO3 and its transcriptional target glycine-N-methyltransferase (GNMT) whose upregulation triggers depletion of s-adenosylmethionine (SAM), a key co-substrate involved in methyl group transfer reactions. Mechanistically, we demonstrate that reduced intracellular SAM availability disrupts carboxymethylation and maturation of nuclear lamin, which induce cytosolic release of chromatin fragments and subsequent activation of the cGAS/STING-IFN-I cascade to suppress neurogenic differentiation. Together, our findings suggest the FOXO3-GNMT/SAM-lamin-cGAS/STING-IFN-I signaling cascade as a critical stress response program that regulates long-term regenerative potential.

[1] Department of Pathology and Laboratory medicine, Weil Cornell Medicine, New York, NY 10021, USA. [2] Department of Pharmacology and Cancer Biology, Duke University School of Medicine, Durham, NC 27710, USA. [3] Department of Neurosurgery, Southwest Hospital, Chongqing 400038, China. [4] Feil Family Brain and Mind Research Institute, Weil Cornell Medicine, New York, NY 10021, USA. [5] Department of Medicine, Weil Cornell Medicine, New York, NY 10021, USA. [6] Meyer Cancer Center, Weill Cornell Medicine and New York Presbyterian Hospital, New York, NY 10021, USA. ✉email: jep2025@med.cornell.edu

Stem cells persist along the mammalian lifespan to maintain tissue homeostasis by replacing damaged or lost cells[1]. Meanwhile, elevation of stress response pathways, a hallmark of aging tissues, acts to promote adult stem cell depletion through induction of senescence or cell death[2–4]. Intrinsic and extrinsic cell-stressors such as oxidative stress, DNA damage, mitochondrial dysfunction, loss of proteostasis, and the inflammatory tissue milieu are the well-known factors that contribute to increased stress response. Among them, oxidative insults inflict damage to cellular macromolecules that lead to cytostasis, cytotoxicity and ultimately, the functional decline of stem cells[1]. Studies in various model organisms have identified a range of key stress response molecules (i.e., Atm[5], Foxo3[6–10], Prdm16[11]) that also function to protect stem cell reserves against physiological and pathological oxidative insults. However, the molecular pathway(s) that translate the stress signals to the cellular behaviors remain poorly understood. Forkhead box protein O (FOXO) transcription factors play evolutionarily conserved roles in a wide range of biological processes from aging to metabolism, not only by sensing stress but also through promoting stress resistance[12]. For example, previous studies indicated that FOXO is required for long-term regenerative potential of the hematopoietic stem cell by regulating the response to physiologic oxidative stress and quiescence[7]. Notably, among the many organs, the brain is particularly vulnerable to the oxidative stress due to its high oxygen consumption, unusual enrichment of polyunsaturated fatty acids as well as the presence of excitotoxic amino acids[13]. As a result, neural stem/progenitor cells (NSPCs) and their functional progenies in adult brains face constant stressful challenges that affect their neurogenic potential[14,15]. Indeed, previous studies found that FOXO expression in the central nervous system not only serves a key role in preserving neural stem cell pools[9,10], it also protects neurons against age-related axonal degeneration across species[16–18]. But despite these revelations, there still lacks a mechanistic understanding of how oxidative stress affects FOXO activation systematically and whether and how that contributes to the neuroprotective responses.

The type-I interferon (IFN-I) response is an innate immune response that can be induced by a number of pattern recognition receptors[19]. Among them, cytosolic DNA fragments are recognized by cyclic GMP-AMP synthetase (cGAS), which initiates reaction of GTP and ATP to form cyclic GMP-AMP (cGAMP), a ligand of the signaling adapter stimulator of interferon genes (STING, TMEM173). The binding of cGAMP to STING activates TANK Binding Kinase 1 (TBK1) kinases-mediated phosphorylation of transcription factor IRF3 that triggers IFNα/β production and subsequent IFN response[20–23]. Increased IFN-I response has been shown to promote NSPC quiescence and suppress neurogenic differentiation[14,24]. Interestingly, a recent study revealed that IFN-I signaling is elevated in the brain of aged humans and animals and correlates with increased oxidative stress[14]. But the connection between oxidative stress and IFN-I response is unclear.

Here, we report that oxidative stress-induced FOXO3 activation promotes transcriptional upregulation of glycine-N-methyltransferase (GNMT) that triggers intracellular depletion of S-adenosyl-L-methionine (SAM). Using NSPCs as a system, we further demonstrated that reduction of intracellular SAM availability compromises nuclear lamin maturation that would eventually lead to cytosolic DNA leakage, cGAS/STING activation, and IFN-I response to suppress neurogenic differentiation. These findings established FOXO3-GNMT/SAM-lamin-cGAS/STING-IFN-I signaling cascade as a critical stress response program that regulates NSPC differentiation.

## Results

**High redox potential-mediated cellular stress activates IFN-I pathway**. The neurogenic differentiation potential of NSPCs declines under iatrogenic insults, traumatic injuries, or inflammatory stress conditions[14,25]. To determine how oxidative stress signal impacts NSPC differentiation, we subjected the cultured murine NSPCs to either pro-oxidant agent paraquat (PQ) or anti-oxidant N-acetylcysteine (NAC). Measurement by a ratiometric Grx-roGFP2 sensor confirmed that PQ treatment induced a marked elevation of intracellular redox potential relative to the mock-treated NSPCs, whereas NAC treatment led to a significant reduction of redox potential (Fig. 1a, b). It is worthy to note that the PQ dosages used in the study were not cytotoxic in NSPCs (Supplementary Fig. 1a). Compared to the mock-treated control NSPCs, we found that NSPCs under PQ but not NAC treatment, exhibited marked reduction in their production of TUBB3-positive newly born neurons when induced to differentiate (Fig. 1c), suggesting a regulatory role of oxidative stress response on neurogenic differentiation.

To determine the signaling pathway that underlies the oxidative stress-induced neurogenic decline, we next performed gene expression profiling against mock-treated control and NSPCs following 48 h redox preconditioning. Gene set enrichment analysis of differentially regulated genes revealed IFN-I signaling as one of the most enriched signature pathways following PQ treatment (Fig. 1d; Supplementary Data 1). Concordantly, heatmap visualization showed an overall increase of IFN-I pathway gene expression pattern in PQ-treated NSPCs, as compared to the control or NAC-treated cells (Fig. 1e). Quantitative real-time PCR (qRT-PCR) confirmed the transcriptional upregulation of major IFN-I signaling downstream surrogates, including Ifnb1, Isg15, Socs1, and Mx1 (Fig. 1f). Importantly, the PQ-induced IFN-I signaling gene upregulation was completely blocked by co-treatment of anti-oxidant NAC, indicating that the response is redox potential dependent. Consistently, ELISA analysis of PQ-treated NSPCs revealed significantly elevated secretion of the key IFN-I response effector IFNβ as compared to the mock-treated control cells (Fig. 1g). Similar results were obtained in $H_2O_2$ treated NSPCs (Supplementary Fig. 1b, c).

To test whether activation of IFN-I signaling accounts for the redox stress-induced neurogenic decline, we next directly treated the NSPCs with IFNβ. Indeed, addition of IFNβ alone was sufficient to suppress neurogenic differentiation of NSPCs (Fig. 1h). Conversely, depletion of interferon-α/β receptor (IFNAR) β chain (IFNAR2), a key subunit of IFNAR dimer, blocked IFN-I response in PQ-treated NSPCs and restored their neurogenic potential (Supplementary Fig. 1d, e; Fig. 1i). These data collectively suggest that oxidative stress signaling regulates neurogenic differentiation of NSPCs through activation of IFN-I pathway.

**FOXO3 is required for ROS-induced IFN-I response**. FOXO proteins are known to regulate physiological oxidative stress response partly due to their role in modulating the transcriptional expression of ROS-scavenging enzymes[26,27]. Indeed, heatmap and GSEA analysis confirmed that transcriptional targets of FOXO3 were upregulated in PQ-treated NSPCs (Supplementary Fig. 2a, b). To determine the role of FOXO3 in ROS-induced IFN-I response, we next analyzed how FOXO3 depletion affects oxidative stress-induced IFN-I signaling activation. As expected, PQ treatment to the control NSPCs (non-targeted guide RNA, sg-NT) elicited a robust IFN-I response, as evidenced by the markedly enhanced phosphorylation of IFN-I upstream and

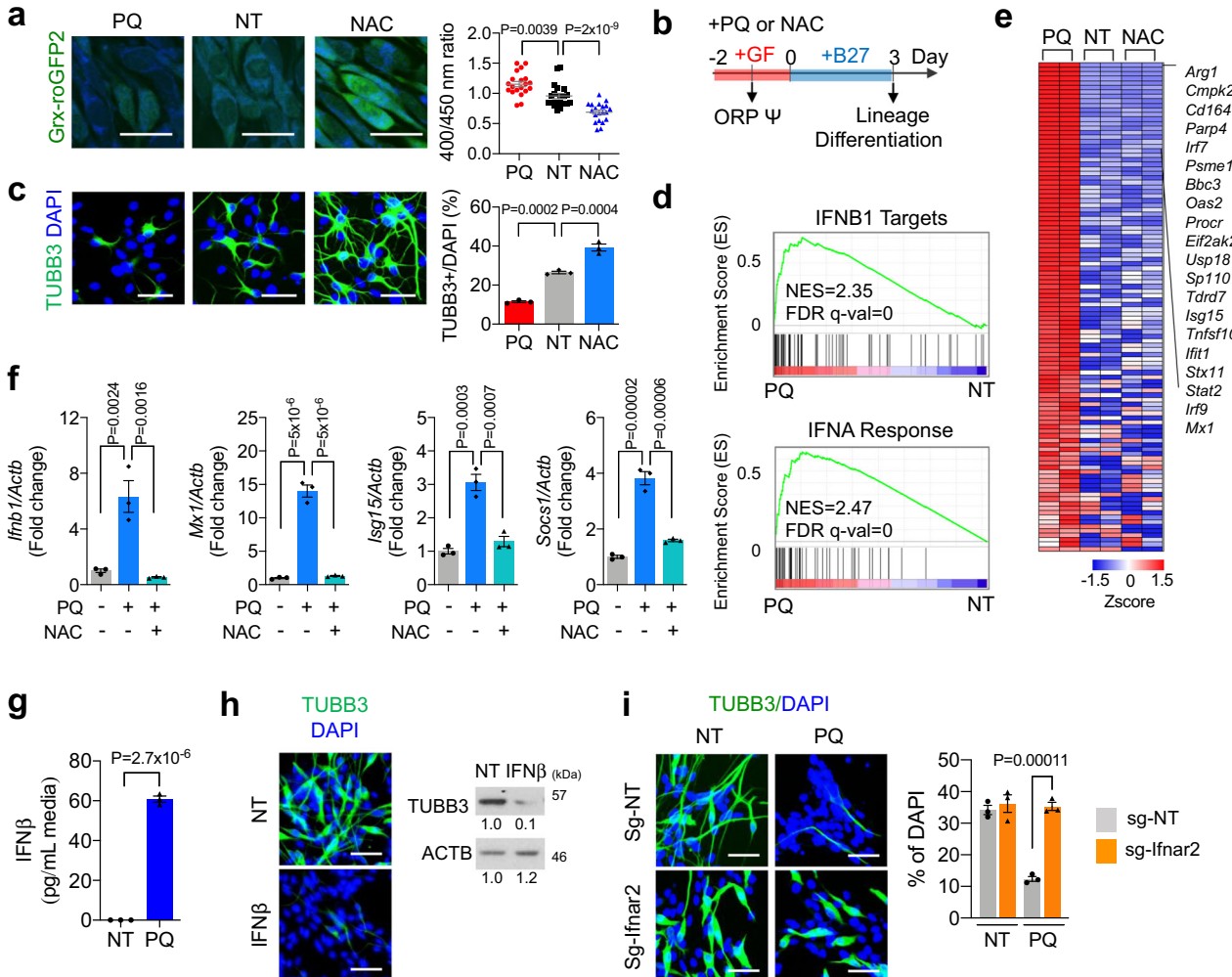

**Fig. 1 Cellular stress response under high redox potential activates IFN-I responses. a** Left, representative images of treated Grx1-roGFP2-NSPC. Right, quantitation for redox potential of Grx1-roGFP2-NSPC following 24 h treatment. Scale bar = 20 μm. Mean ± s.e.m. of 20 cells. PQ: 5 μM of paraquat, NT: mock-treated control, NAC: 5 mM of N-acetylcysteine. **b** Schema for treatment of NSPC culture. ORP: oxidation/reduction potential. **c** Representative immunofluorescence analysis (IF) images of TUBB3 on 3 day of differentiation. Scale bar = 50 μm. Percent of TUBB3-positive cells is plotted on the right. Mean ± s.e.m. of three independent experiments. **d** One-sided GSEA analysis of PQ-upregulated genes showed significant enrichment of 'Hecker_IFNB1_Targets' (upper) and 'Hallmark_IFNA_response' (lower) gene sets in NSPCs treated with or without PQ. **e** Heatmap visualization of 108 differentially expressed genes (complete list is available as source data) related to IFN-I response from NSPCs treated with PQ (5 μM) or NAC (5 mM) for 2 days. Top 20 genes are presented on the right. The red represents a positive Z-score (upregulated), and the blue represents a negative one (downregulated). **f** qRT-PCR results for IFN-1 stimulated genes (ISGs) in NSPCs following 4 days of PQ and NAC treatment. Mean ± s.e.m. of three independent experiments. **g** IFNβ secretion in the media following 48 h treatment. Mean ± s.e.m. of three independent experiments. **h** Representative IF (left) and western blot analysis (WB, right) results of NSPCs differentiated following 2 days of IFNβ (40 ng/mL) treatment. Scale bar = 50 μm. **i** Representative IF images of TUBB3 in NSPCs expressing non-targeted guide RNA (sg-NT) or Ifnar2 targeted guide RNA (sg-Ifnar2) following 3 days of differentiation. Scale bar = 50 μm. Percent of TUBB3-positive cells is plotted on the right. Mean ± s.e.m. of three independent experiments. Statistical significance was determined by one-way ANOVA for **a**, **c**, **f**, and **i**, and by two-sided unpaired *t*-test for (**g**). Experiments for **a**, **c**, **h**, and **i** were repeated three times independently with similar results and representative images/blots are shown.

downstream signaling protein STAT1 and TBK1 (Fig. 2a, b), elevated IFNβ secretion (Fig. 2c), and the strongly upregulated mRNA expression of IFN-I stimulated genes (ISGs) (*Isg15*, *Socs1*, *Usp18* and *Nos2*) (Fig. 2d). In comparison, the PQ treatment-induced IFN-I response and activation of its upstream and downstream signaling were evidently attenuated in FOXO3-depleted NSPCs (sg-Foxo3) (Fig. 2b–d), suggesting that FOXO3 plays a crucial role in regulation of ROS-induced IFN-I response. Interestingly, FOXO3 depletion in NSPCs also attenuated IFN-I response induced by other stressors (i.e., $H_2O_2$, oligomycin, zeocin) (Supplementary Fig. 2c–e), suggesting that FOXO3 may

serve as a general cellular stress effector by activating IFN-I response.

FOXO3 integrates a variety of cellular signals that modulate its transcriptional activity[28]. To examine whether activation of FOXO3 by itself was sufficient to trigger IFN-I response independently oxidative stress, we transduced NSPCs with an adenoviral-encoded activated mutant form of FOXO3 (FOXO3[TA], triple alanine form[29]) that was known to localize to the nucleus (Fig. 2e, f). FOXO3[TA] expression markedly increased IFNβ secretion as well as enhanced expression of ISGs as compared to the control adenovirus-infected NSPCs (Fig. 2g, h),

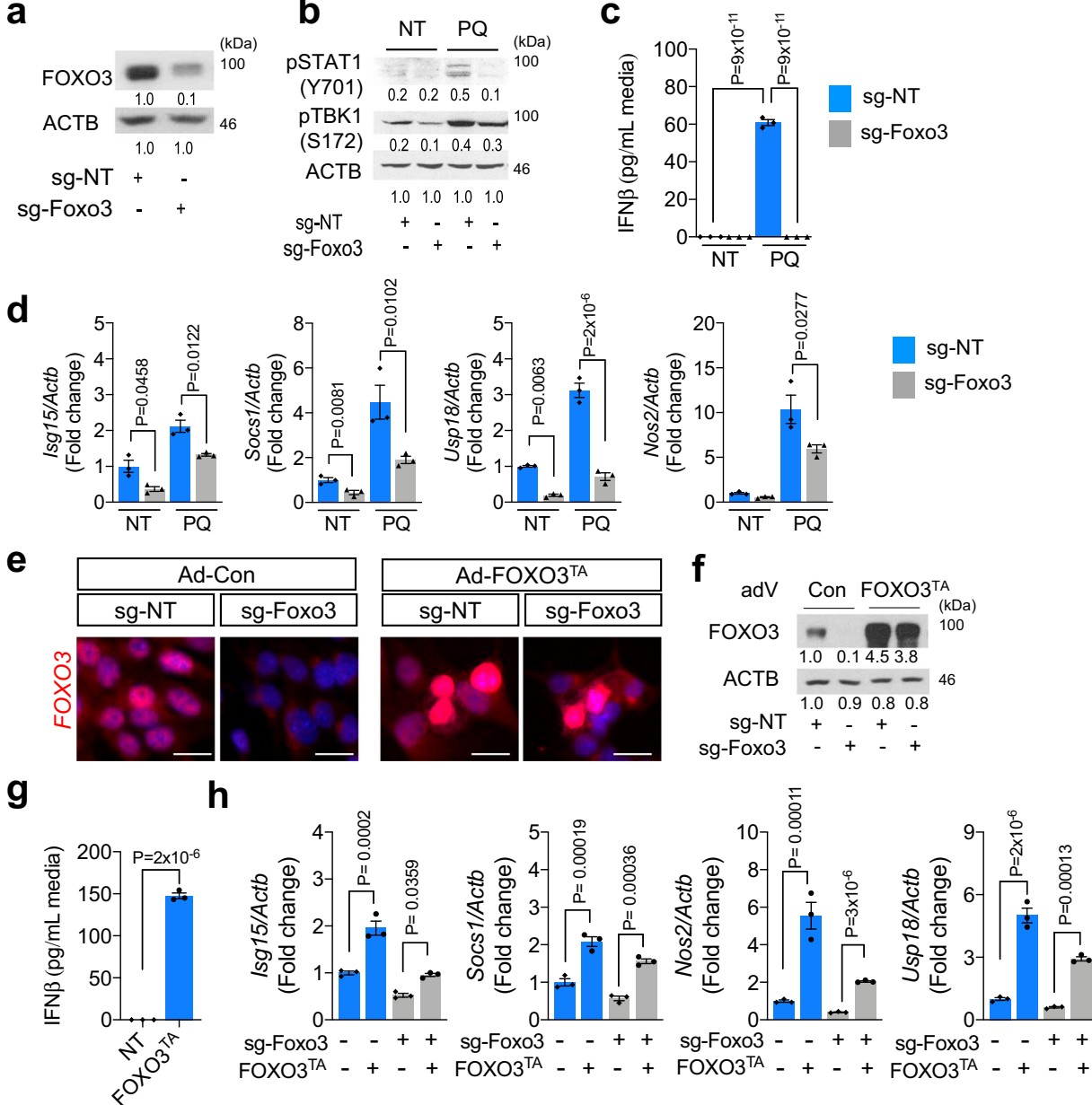

**Fig. 2 FOXO3 is necessary for ROS-induced IFN-I response. a** WB for FOXO3 in NSPCs expressing non-targeted guide RNA (sg-NT) or Foxo3 targeted guide RNA (sg-Foxo3). **b** WB for STAT1 and TBK1 phosphorylation following 48 h treatment. NT: mock-treated control, PQ: 5 μM of paraquat. **c** IFNβ secretion in the media following 48 h treatment. Mean ± s.e.m. of three independent experiments. **d** qRT-PCR results for ISGs following 4 days of PQ treatment. Mean ± s.e.m. of three independent experiments. Representative IF (**e**) and WB (**f**) results for FOXO3 on either adenovirus for control or FOXO3^TA-infected NSPCs. Scale bar = 10 μm. **g** IFNβ secretion in the media following 72 h of control or FOXO3^TA adenovirus infection. Mean ± s.e.m. of three independent experiments. **h** qRT-PCR results for ISGs at 4 days after the infection of adenovirus. Mean ± s.e.m. of three independent experiments. Statistical significance was determined by one-way ANOVA for (**c**, **d**, and **h**), and by two-sided unpaired *t*-test for **g** Experiments for **a**, **b**, **e**, and **f** were repeated three times independently with similar results and representative images/blots are shown.

indicating that FOXO3 is directly responsible for oxidation stress-induced IFN-I activation.

**Oxidation of FOXO3 activates IFN-I response**. Previous reports suggest that ROS signaling activates FOXO by inducing its nuclear translocation[30,31]. Similarly, we found that ROS treatment to NSPCs stimulated FOXO3 nuclear retention and activation that concomitantly led to an elevated FOXO3 protein expression (Fig. 3a–c). Conversely, treatment with the antioxidant NAC not only promoted FOXO3 cytoplasmic shuttling and reduced its expression as well as transcriptional activity, it

also prevented the ROS-induced FOXO3 protein accumulation in PQ-treated NSPCs (Fig. 3a–c, Supplementary Fig. 3a).

The nucleo-cytoplasmic shuttling of FOXO is controlled through a combination of posttranslational modifications, particularly AKT-mediated phosphorylation that promotes its cytoplasmic sequestering[28]. Notably, previous studies reported that ROS treatment could cause a significant reduction of FOXO phosphorylation at threonine 32/serine 253, suggesting that oxidative stress may induce FOXO3 nuclear translocation by impeding its phosphorylation[32,33]. Since reversible cysteine thiol oxidation is a well-known mechanism that regulates signaling

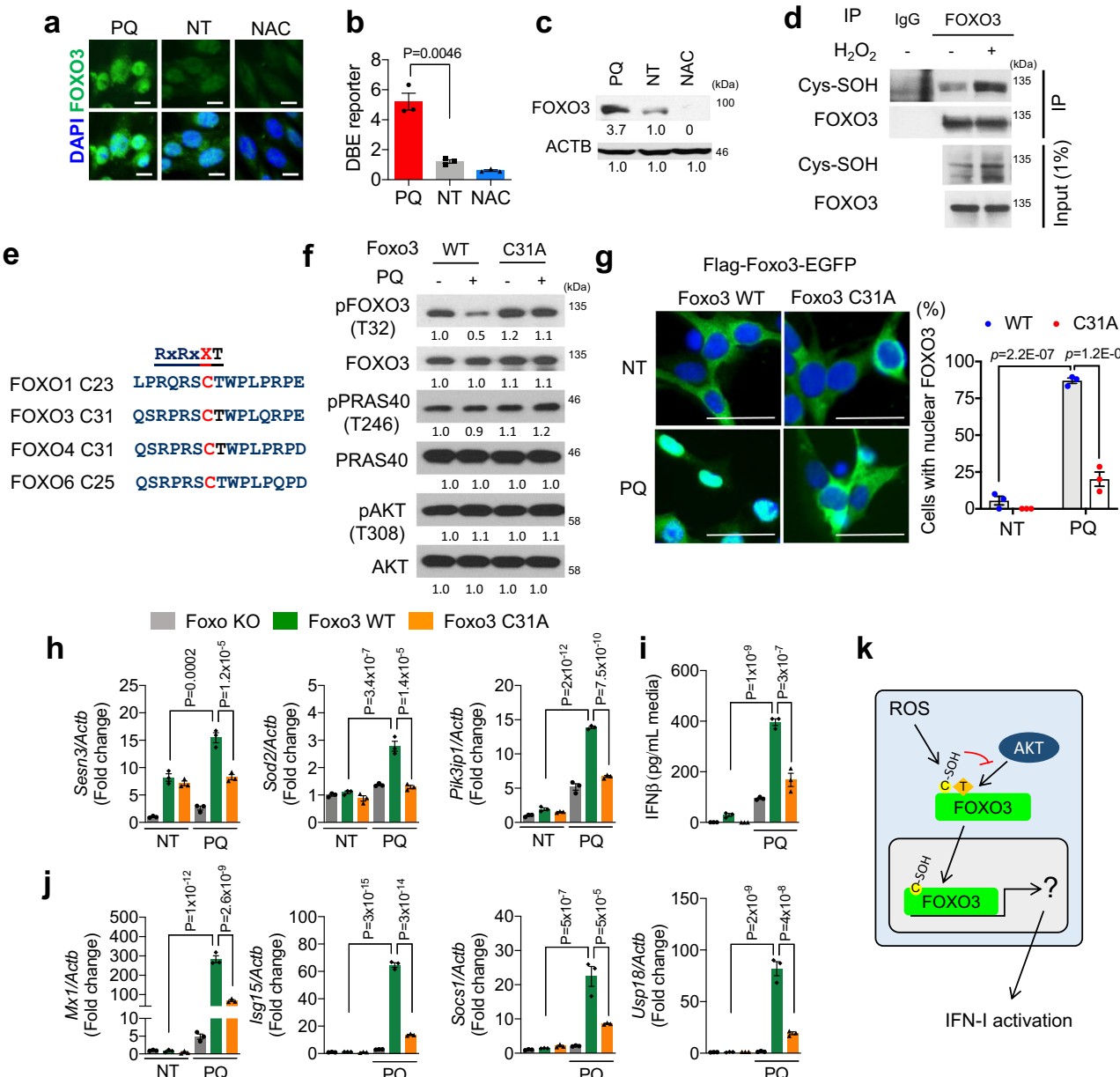

**Fig. 3 Oxidation at Cys31 of FOXO3 activates IFN-I response.** IF (**a**), DBE reporter (**b**, mean ± s.e.m., $n = 3$ independent experiments), and WB (**c**) for FOXO3 in NSPCs following 24 h of respective treatment. Scale bar = 10 μm. **d** WB for cysteine sulfenylation (Cys-SOH) following immunoprecipitation of FOXO3. **e** Conserved consensus sequence adjacent to AKT phosphorylation site of mouse FOXO proteins. **f** WB for indicated proteins from non-treated and PQ-treated (40 μM, 0.5 h) Foxo3 WT or C31A mutant transduced NSPCs. **g** Microscopic analysis of Foxo3 WT or C31A mutant tagged with c-terminus EGFP with or without PQ treatment (40 μM, 16 h). Scale bar = 20 μm. Percent of cells with nuclear FOXO3-EGFP is plotted on the right. Mean ± s. e.m. of three independent experiments. qRT-PCR analysis for transcriptional targets of FOXO3 (**h**) and ISGs (**j**). Foxo null NSPCs with WT or C31A mutant Foxo3 were analyzed following 4 days of PQ treatment. Mean ± s.e.m. of three independent experiments. **i** IFNβ secretion in the media following 48 h treatment. Mean ± s.e.m. of three independent experiments. **k** Schema for activation of FOXO3 by oxidation at Cys31 residue. For **b**, **g**, **h**, **i**, and **j**, statistical significance was determined by one-way ANOVA. Experiments for **a**, **c**, **d**, **f**, and **g** were repeated three times independently with similar results and representative images/blots are shown.

cascades and protein activities[34], we asked whether redox stress conditions could also induce cysteine oxidation in FOXO3. Indeed, immunoprecipitation followed by blotting analysis against cysteine sulfenic acid (Cys-SOH) confirmed a strong elevation of FOXO3 sulfenylation in $H_2O_2$ or PQ-treated NSPCs, as compared to their mock-treated controls (Fig. 3d; Supplementary Fig. 3b). Mammalian FOXO3 contains a highly conserved Cys residue (Cys31) adjacent to threonine (Thr) that is subjected to AKT phosphorylation (Fig. 3e). To test whether the Cys31

oxidation affected AKT-dependent Thr32 phosphorylation, we reconstituted the FOXO3-null NSPCs with a lentiviral construct encoding green fluorescent protein (GFP)-tagged either wild-type (WT) or Cys31 to alanine FOXO3 mutant (C31A). Immunoblot analysis showed that ROS treatment strongly reduced Thr32 phosphorylation of FOXO3 WT but not that of the C31A FOXO3 mutant, as relative to the mock-treated control cells (Fig. 3f), suggesting that oxidation at Cys31 may impede Thr32 phosphorylation. In line with this, fluorescence live-imaging of GFP-tagged

FOXO3 indicated that the ROS-induced nuclear translocation was significantly lower for C31A mutant compared to the wild-type FOXO3 (Fig. 3g). Consistently, analysis of ROS-treated NSPCs transduced with C31A mutant revealed an attenuated induction of FOXO3 downstream genes (i.e., Sod2, Sesn3, pik3ip1), ISGs expression, and IFNβ secretion, as relative to the wild-type FOXO3-transduced cells (Fig. 3h–j). Notably, although the C31A mutant was defective of ROS-induced FOXO3 nuclear shuttling and activation, this mutation did not affect FOXO3 nuclear translocation upon treatment with either PI3K inhibitor (GDC0941) or AKT inhibitor (MK2206) (Supplementary Fig. 3c–e). These findings suggest that Cys thiol oxidation and its associated inhibitory function on FOXO3 phosphorylation is a key mechanism that underlies ROS-regulated FOXO3 and its downstream signaling activation (Fig. 3k).

**Compromised lamin processing upon oxidative stress invokes IFN-I response.** The IFN-I signaling is a cellular innate immune response and often triggered by the cytosolic DNA-sensing cGAS/STING pathway[35]. To examine whether ROS-induced IFN-I activation is mediated by aberrant cytosolic DNA appearance, we treated the cGAS-GFP-expressing NSPCs with pro-oxidant PQ. Fluorescence microscopic analysis of the PQ-treated cells revealed an increased nuclear leakage as evidenced by the emergence of cytosolic H3K4m3-positive chromatin and the inappropriate formation of cytoplasmic cGAS-GFP-containing DNA foci (Fig. 4a), suggesting a compromised nuclear envelope integrity.

The nuclear lamina is essential for the maintenance of nuclear shape and mechanics, and its dysregulation causes nuclear envelopathies and accumulation of cytosolic chromatin fragments[36]. As an essential component of nuclear lamina, the maturation of functional lamin A/B from newly synthesized prelamin A/B follows a multistep process of posttranslational modification that involves farnesylation and methylation of its C-terminal cysteine before proteolytic cleavage of its C-terminal 15 amino acids (Fig. 4b). To test whether oxidative stress affects lamin distribution, we stably transduced NSPCs with a construct encoding either N-terminal GFP-tagged prelamin A (GFP-LMNA) or mCherry-tagged prelamin B1 (mCh-LMNB1). Strikingly, we found that compared to the control cells in which the tagged lamin proteins dispersed evenly along the nuclear envelopes, a large portion of PQ-treated NSPCs displayed an irregular lamin distribution, reminiscent of protein aggregation (Fig. 4c). To test whether the disruption of lamin processing could activate IFN-I response upon oxidative stress, we stably expressed in NSPCs a cysteine 585 to serine prelamin B1 mutant (LMNB1$^{CS}$) that is defective of prelamin maturation-essential farnesylation and methylation[37]. Immunofluorescence analysis of the cells indicated that the LMNB1$^{CS}$ mutant protein, unrecognizable by the mature lamin B1-specific 8D1 monoclonal antibody[38], formed the aggregate-like nucleoplasmic foci that were similar to the ones observed in wild-type lamin B1-transduced NSPCs under ROS treatment (Fig. 4d). qRT-PCR analysis further revealed that expression of LMNB1$^{CS}$ mutant alone was able to induce IFN-I response and downstream gene expression, and the effect could be further enhanced by ROS treatment (Fig. 4e). Conversely, expression of a C-terminus deletion form of mature lamin A mutant (LMNA$^m$) strongly attenuated the ROS-induced IFN-I signaling activation and ISGs expression (Fig. 4f, g). These findings together suggest defective lamin processing as an underlying cause of IFN-I response under oxidative stress.

**ROS-induced intracellular SAM depletion disrupts lamin maturation.** Lamin maturation requires isoprenylation and methylation on the C-terminal cysteine residues[39]. To determine how ROS regulates lamin posttranslational modification, we performed targeted quantitative polar metabolomics profiling by liquid chromatography-tandem mass spectrometry (LC–MS) on samples derived from control, pro- or anti-oxidant treated NSPCs. Among the 258 metabolites analyzed, we found that the turnover of SAM exhibited an inverse correlation with redox potential. As compared to the mock-treated control cells, treatment with the pro-oxidant PQ gave rise to a 1.5-fold reduction of cellular SAM and a 2.1-fold reduction of SAM to SAH ratio (Fig. 5a). By contrast, NSPCs treated with anti-oxidant NAC exhibited a 1.4-fold increase of cellular SAM level.

SAM is a principal methyl donor for a variety of biological processes including isoprenylcysteine carboxymethyl-transferase (ICMT)-mediated lamin methylation (Fig. 5b). Since prelamin methylation is a prerequisite step of lamin maturation, we next examined the effect of SAM depletion on lamin processing by treating NSPCs with cycloleucine (CL), a methionine adenosyl-transferase 2A (MAT2A) inhibitor. Immunofluorescence and immunoblot analyses revealed that inhibition of SAM production compromised lamin maturation, as indicated by the significantly reduced levels of 8D1-positive mature lamin B1 in CL- or PQ-treated NSPCs compared to their mock controls (Fig. 5c, d). In addition, CL treatment alone was sufficient to elicit IFNβ secretion and induction of ISGs (Fig. 5e, f). Consistently, disrupting lamin methylation by depletion of its methyltransferase ICMT (sg-Icmt) also inhibited lamin B1 maturation and provoked IFN-I response, as evidenced by the induction of TBK1 phosphorylation (Fig. 5g; Supplementary Fig.4), IFNβ secretion (Fig. 5h), and ISGs expression (Fig. 5i). Together, these findings suggest that SAM depletion and its dependent lamin methylation disruption are the underlying cause of ROS-induced cGAS/STING-IFN-I signaling activation.

**ROS regulates intracellular SAM through GNMT.** SAM is a universal co-substrate of methyl group transfer reactions[40]. Intracellular SAM levels are balanced by MAT2A-catalyzed synthesis and its consumption through multiple catabolic processes[41]. Since our metabolite profiling revealed little change of intracellular methionine— the precursor for SAM (Fig. 5a), we next examined the expression of the major enzymes involved in SAM metabolism. qRT-PCR analysis of control and ROS-treated NSPCs indicated that cellular expression of MAT2A, the enzyme that catalyzes the synthesis of SAM from methionine, remained relatively stable (Supplementary Fig. 5a). By contrast, the expression of GNMT, a key catabolic enzyme that catalyzes the SAM to SAH conversion, was markedly induced by PQ treatment but suppressed by anti-oxidant NAC (Fig. 6a–c).

GNMT catalyzes the reaction of glycine to sarcosine by using SAM as the methyl donor[42]. Likewise, CRISPR/Cas9-mediated GNMT depletion by guide RNA (sg-Gnmt) led to two-fold enhancement of cellular SAM accumulation compared to the control sgRNA transduced cells (Fig. 6d, e). Conversely, doxycycline (DOX)-induced overexpression of exogenous GNMT conferred a rapid ~70% SAM depletion within 24 h and led to a reduction of global H3K4 methylation (Fig. 6f, g), consistent with its role as a principle methyl donor for histone methylation[43,44]. Importantly, DOX-induced GNMT expression-induced IFNβ secretion and initiated a time-dependent IFN-I response by activating its downstream signaling and gene expression (Fig. 6h–j). These findings suggest that GNMT-regulated SAM depletion is a likely route to ROS-induced IFN-I activation.

To determine whether GNMT-regulated SAM depletion could also instigate nuclear leakage to activate cGAS/STING signaling, we transduced the DOX-inducible GNMT expressing NSPCs

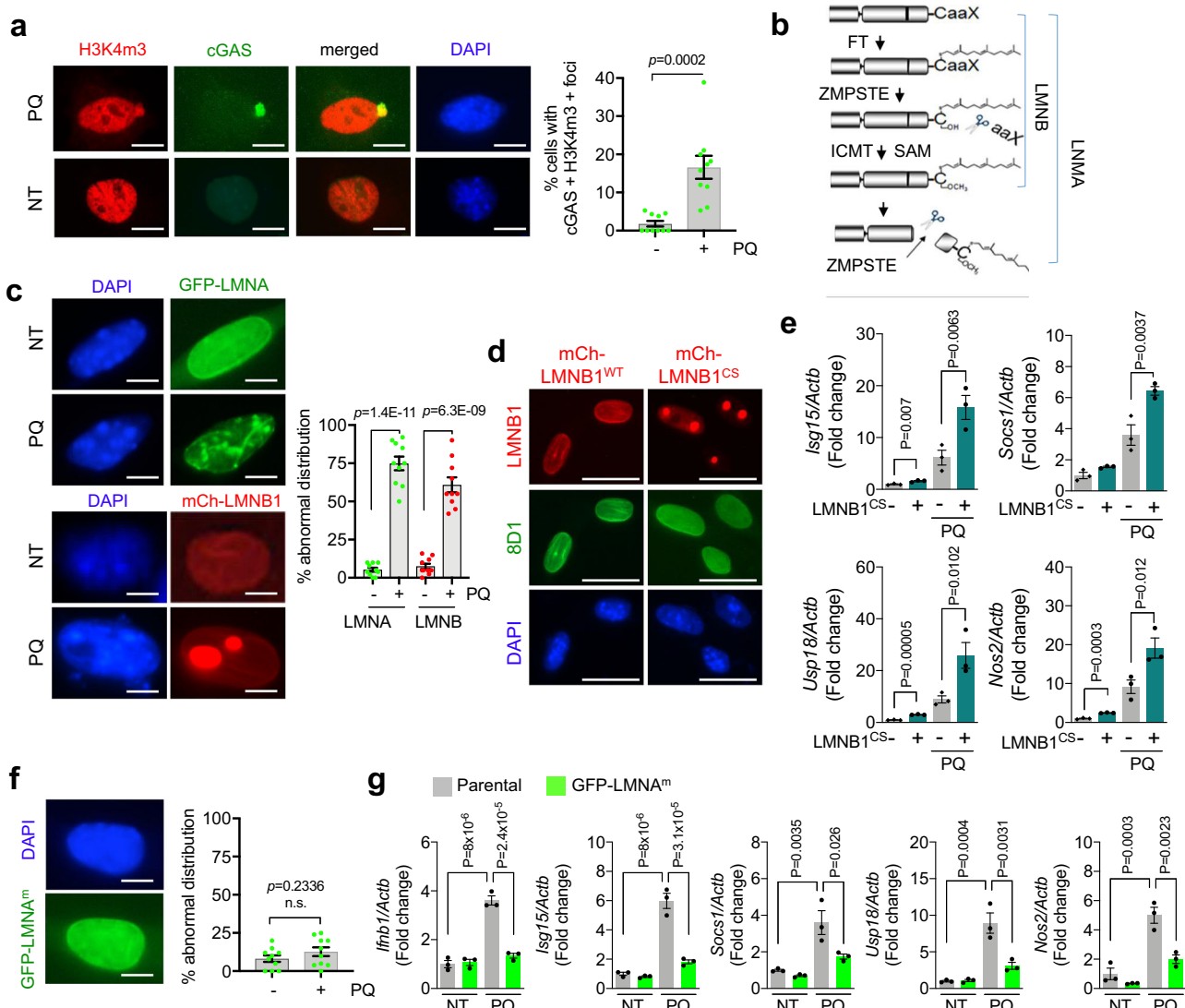

**Fig. 4 Defective nuclear lamin processing causes nuclear leakage under oxidative stress. a** Left, analysis for cGAS-GFP reporter and IF of H3K4m3 following 24 h of PQ treatment. Right, quantitation of the percent of cells with cGAS-GFP foci overlapped cytosolic H3K4m3. Scale bar = 4 μm. Mean ± s.e. m. of ten images. **b** Diagram for lamin processing. FT: Farnesyl transferase, ZMPSTE: ZMPSTE24 protease, ICMT: Isoprenylcysteine carboxymethyltransferase, SAM: S-adenosyl-L-methionine. **c** Left, microscopic analysis for GFP-LMNA and mCherry-LMNB1 (mCh-LMNB1) localization following 24 h of PQ treatment. Right, quantitation for abnormal lamin structure ratio to whole cells. Scale bar = 2 μm. Mean ± s.e.m. of ten images. **d** IF for mature LMNB1 (8D1, green) in NSPCs expressing mCherry-tagged prelamin B1 (mCh-LMNB1^WT) and mCherry-tagged mutant prelamin B1 (mCh-LMNB1^CS). Scale bar = 5 μm. **e** qRT-PCR analysis for ISGs after 4 days of PQ treatment. Mean ± s.e.m. of three independent experiments. **f** Left, microscopic analysis of NSPC expressing mature LMNA (GFP-LMNA^m). Scale bar = 2 μm. Right, quantitation of GFP-LMNA^m is plotted following 24 h of PQ treatment. Mean ± s.e.m. of ten images. **g** qRT-PCR results for ISGs after 4 day of PQ treatment. Mean ± s.e.m. of three independent experiments. Statistical significance was determined by two-sided unpaired t-test for **a**, **c**, and **f** and by one-way ANOVA for **e** and **g**. Experiments for **a**, **c**, **d**, and **f** were repeated three times independently with similar results and representative images/blots are shown.

with a cytosolic DNA fragment-sensing cGAS-GFP construct. Compared to the mock-treated control cells, DOX-treated NSPCs exhibited a significant elevation of cGAS-GFP-containing foci that colocalized with cytosolic H3K4m3-positive DNA fragments (24.4 ± 1.275% vs 0.7 ± 0.45%) (Fig. 6k). Immunoblot analysis of the DOX-treated NSPCs further revealed a strong reduction of 8D1-positive mature lamin B1 protein level (Fig. 6l), suggesting that lamin maturation was compromised following GNMT induction. Consistently, transduction of a mature lamin mutant (GFP-LMNA^m) in the DOX-inducible GNMT expressing NSPCs suppressed the GNMT induction-evoked IFNβ secretion as well as ISGs expression (Fig. 6m, n). Furthermore, co-treatment of NSPCs with cGAS inhibitor, RU.521[45], largely suppressed PQ- or

GNMT expression-induced IFN-I activation and restored their neuronal differentiation capacity (Supplementary Fig. 5b–d). Collectively, these data indicate that GNMT is a key regulator of IFN-I response under ROS treatment.

**Redox stress modulates NSPC neurogenic potential through FOXO3-regulated GNMT expression.** Our data indicate that FOXO3 is directly responsible for oxidative stress-induced IFN-I activation. Indeed, the frequency of PQ-induced cGAS-GFP foci was significantly reduced following FOXO3 depletion (Supplementary Fig. 6a, b). Conversely, enforced expression of the active FOXO3^TA mutant enhanced the percentage of irregular nuclei-containing NSPCs that displayed a reduced 8D1 staining

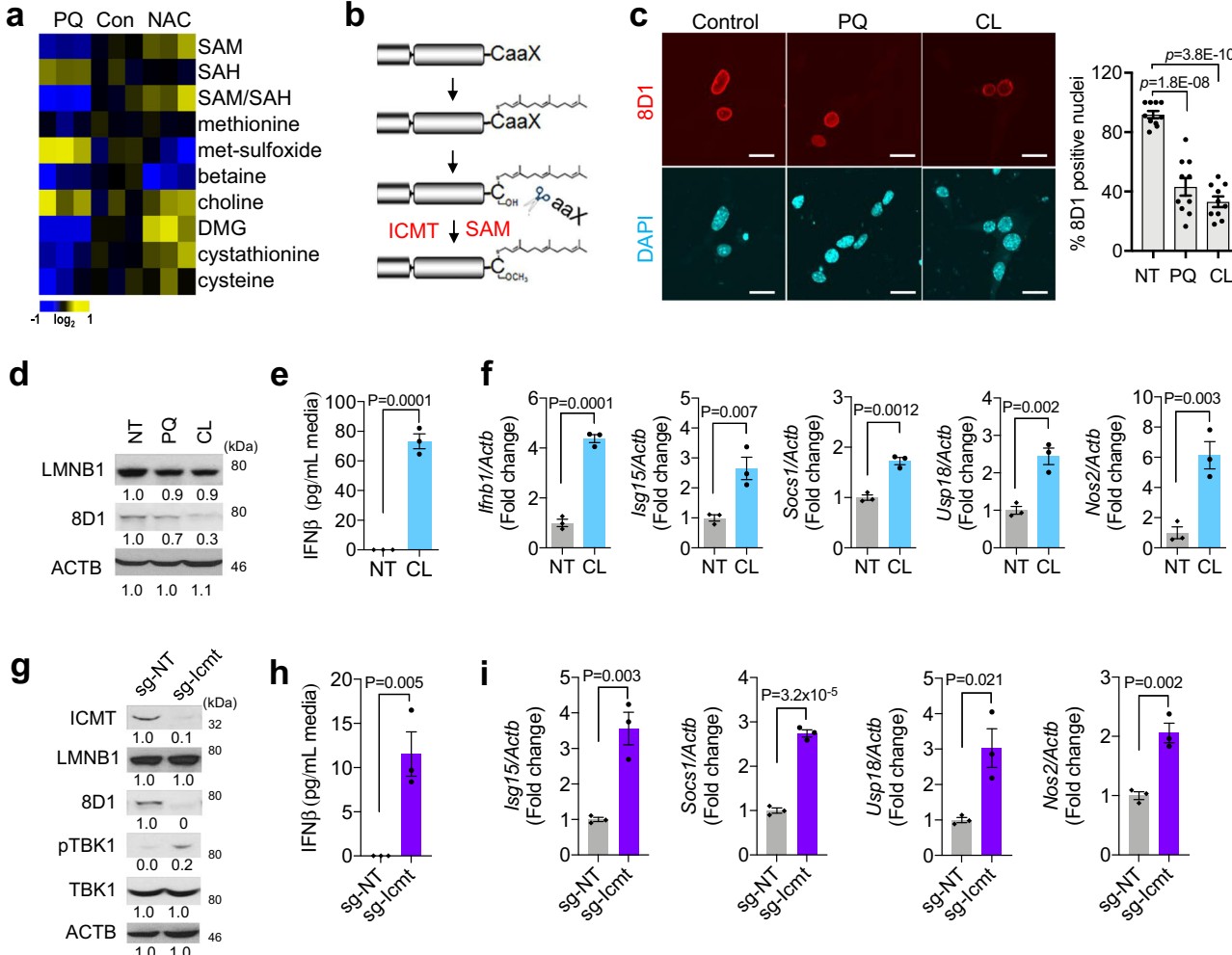

**Fig. 5 Stress-depleted intracellular SAM underlies nuclear leakage through lamin B1 maturation failure. a** Heatmap for metabolites extracted from NSPCs after 2 days of respective treatment. Data are presented as three independent experiments. SAM: S-adenosyl-methionine, SAH: S-adenosyl-homocysteine, DMG: dimethylglycine. **b** Diagram for lamin processing. **c** Left, IF images for mature lamin B1 (8D1, red) following 48 h of PQ or CL treatment. CL: 25 mM of cycloleucine (1-aminocyclopentanecarboxylic acid). Right, quantitation for 8D1-positive nuclei. Scale bar = 10 μm. Mean ± s.e.m. of ten images. **d** WB results for total laminB1 (LMNB1) and mature laminB1 (8D1). IFNβ secretion in the media (**e**) and qRT-PCR analysis for ISGs (**f**) following 48 h of CL treatment. Mean ± s.e.m. of three independent experiments. **g** WB for defect of lamin processing and IFN-I activation in Icmt-targeted guide RNA transduced NSPCs (sg-Icmt). IFNβ secretion in the media (**h**) and qRT-PCR analysis for ISGs (**i**) following 48 h growth. Mean ± s.e.m. of three independent experiments. Statistical significance was determined by one-way ANOVA for (**c**) and by two-sided unpaired t-test for (**e**, **f**, **h**, and **i**). Experiments for **c**, **d**, and **g** were repeated three times independently with similar results and representative images/blots are shown.

(Supplementary Fig. 6c, d), phenocopying GNMT-induced cells. Given our findings that FOXO3 and GNMT were both involved in ROS-induced IFN-I activation, we next investigated the potential connection between FOXO3 activation and GNMT induction under stress conditions. By surveying the ±3 kb genomic DNA sequence near murine GNMT transcription start site (TSS), we identified two putative DAF-16 family-binding element (DBE) FOXO motifs, positioned at 200–300 bp upstream of TSS (Supplementary Fig. 7). Chromatin immunoprecipitation (ChIP) coupled with qRT-PCR revealed that FOXO3 was 9.7 ± 2.7-fold enriched at GNMT promoter relative to the background gene desert (Fig. 7a). Moreover, qRT-PCR analysis confirmed that enforced expression of the active FOXO3[TA] mutant significantly enhanced GNMT transcription, as compared to the control virus infected NSPCs (Fig. 7b). By contrast, CRISPR/Cas9-mediated depletion of endogenous FOXO3 suppressed GNMT mRNA expression (Fig. 7c). These data suggest that FOXO3 transcriptionally controls GNMT expression.

We next examined whether FOXO3-regulated ROS-initiated IFN-I response through GNMT. Treatment of NSPCs with pro-oxidant agent PQ promoted a marked increase of GNMT mRNA and protein expression relative to the mock-treated control cells (Fig. 7c, d). However, this ROS-induced GNMT upregulation was significantly attenuated in the NSPCs depleted of FOXO3. As expected, the FOXO3-depleted NSPCs exhibited a steady increase of cellular SAM levels as relative to their respective controls before or after ROS treatment (Fig. 7e). Concordantly, knockdown of GNMT abolished FOXO3[TA] expression-induced IFN-I response and downstream gene expression (Fig. 7f).

We next went on to determine how FOXO3-GNMT/SAM-IFN-I signaling pathway regulates neurogenesis under oxidative stress condition. As expected, immunoblot and immunofluorescence analysis revealed that treatment of NSPCs with pro-oxidant PQ attenuated their neuronal differentiation capacity, as indicated by the reduction of both the expression level of neuronal marker TUBB3 and the percentage of TUBB3-positive cell population

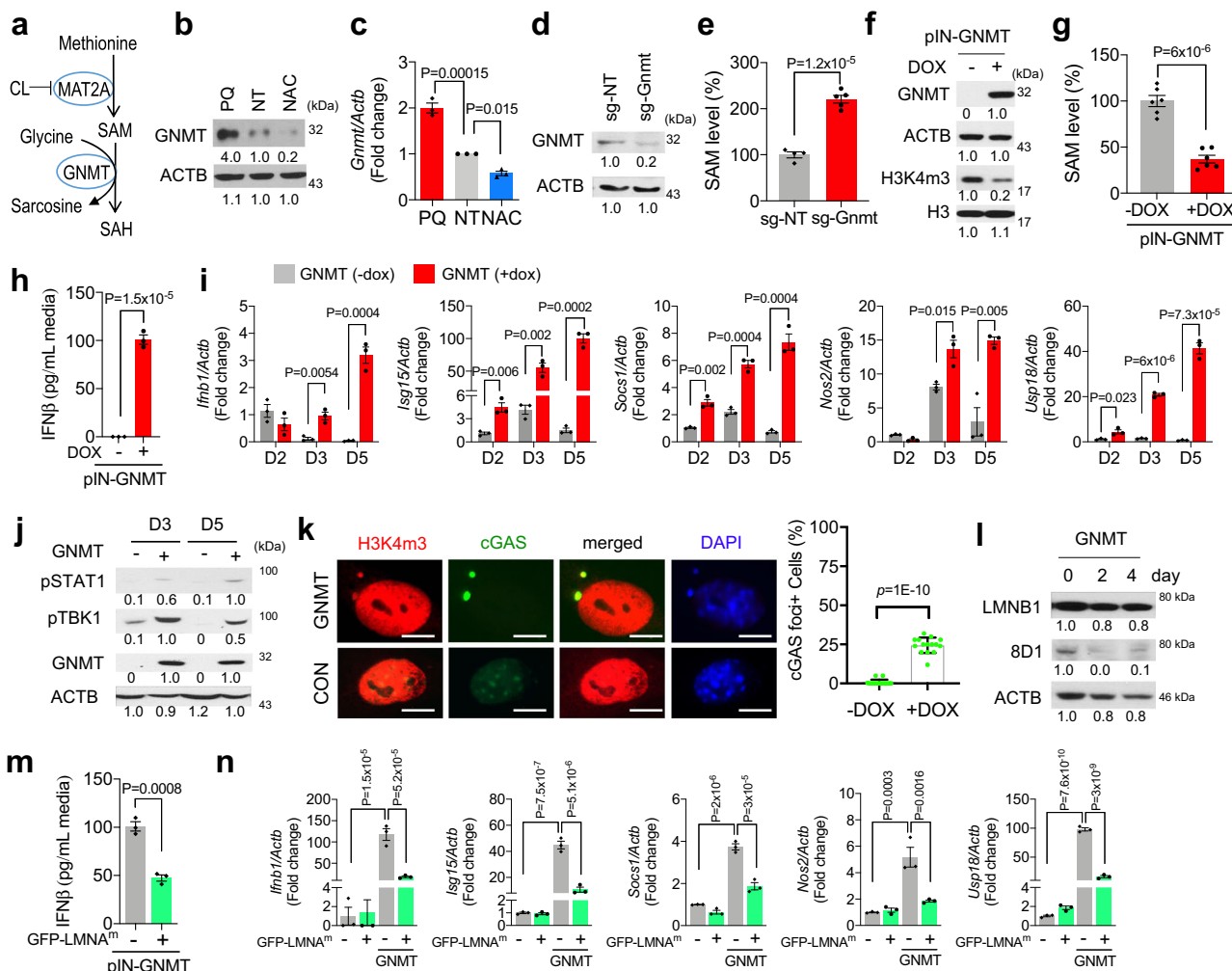

**Fig. 6 Stress-induced GNMT depletes intracellular SAM. a** Diagram for GNMT function. MAT: methionine adenosyltransferase, GNMT: glycine N-methyltransferase. WB (**b**) or qRT-PCR (**c**) analysis for GNMT expression in NSPCs following 2 days of indicated treatment. Mean ± s.e.m. of three independent experiments. **d** WB for GNMT in GNMT-targeted guide RNA-expressing NSPC (sg-Gnmt). **e** SAM levels following 48 h treatment in sg-NT (gray bar, $n = 4$) or sg-Gnmt (red bar, $n = 5$) infected NSPC. 100% of SAM = 50 μM. Data are presented as mean ± s.e.m. WB (**f**), SAM level ($n = 6$) (**g**), and IFNβ secretion in the media ($n = 3$) (**h**) following induction of GNMT with 2 μg/mL of doxycycline (DOX) for 2 days. 100% of SAM = 50 μM. Data are presented as mean ± s.e.m. **i** qRT-PCR results for ISGs at indicated points after induction of GNMT. Mean ± s.e.m. of three independent experiments. **j** WB for STAT1 or TBK phosphorylation following induction of GNMT at indicated points. **k** Left, microscopic analysis of cGAS-GFP reporter and IF of H3K4m3 following induction of GNMT. Scale bar = 2 μm. Right, quantitation of the percent of cells with cGAS-GFP foci overlapped cytosolic H3K4m3. Mean ± s.e.m. of ten images. **l** WB for total lamin B1(LMNB1) and mature lamin B1 (8D1) expression following induction of GNMT. **m** IFNβ secretion in the media following 48 h of GNMT induction in GFP-LMNA^m expressing NSPCs (GFP-LMNA^m). Mean ± s.e.m. of three independent experiments. **n** qRT-PCR for ISGs following 4 days of GNMT induction. Mean ± s.e.m. of three independent experiments. Statistical significance was determined by two-sided unpaired t-test for **e**, **g**, **h**, **i**, **k**, and **m**, and by one-way ANOVA for (**c**, **n**). Experiments for **b**, **d**, **f**, **j**, **k**, and **l** were repeated three times independently with similar results and representative images/blots are shown.

relative to the mock-treated control cells (Fig. 7g–j). Evidently, depletion of either FOXO3 or GNMT in the PQ-treated NSPCs reversed the ROS effect and was sufficient to restore their neurogenic potential (Fig. 7g–j). Notably, the FOXO3 depletion-promoted neuronal differentiation could be further blocked by DOX-induced exogenous GNMT expression, consistent with our finding that GNMT is a downstream effector of FOXO3 signaling (Fig. 7k).

To examine our findings under in vivo context, we next applied a well-established oxidative stress model by performing transient middle cerebral artery occlusion (tMCAO) followed by reperfusion in brain parenchyma[46,47] (Supplementary Fig. 8a, b). Analysis of FOXO3 expression around SVZ regions showed a time-dependent increase of nuclear FOXO3 and loss of 8D1-positive mature lamin expression in NESTIN + progenitor cell population following 1 h tMCAO (Supplementary Fig. 8c, d). Combined immunoblot and qRT-PCR analysis of the micro-dissected SVZ tissues further confirmed the tMCAO-induced increase of FOXO3, GNMT as well as other FOXO3 downstream and ISG gene expression (Supplementary Fig. 8e, f), suggesting ROS-FOXO3-GNMT/SAM-lamin-IFN-I signaling cascade is operative in vivo.

Lastly, considering the elevation of ROS in aging brain, we further examined type-I IFN stimulated gene expression in young and old (<60 year old) and aged (>60 year old) patient brain samples. Consistently, we observed a clear increase of ISGs along with GNMT mRNA expression in aged brains (Supplementary Fig. 9a, b). Altogether, our results suggest the FOXO3-GNMT/

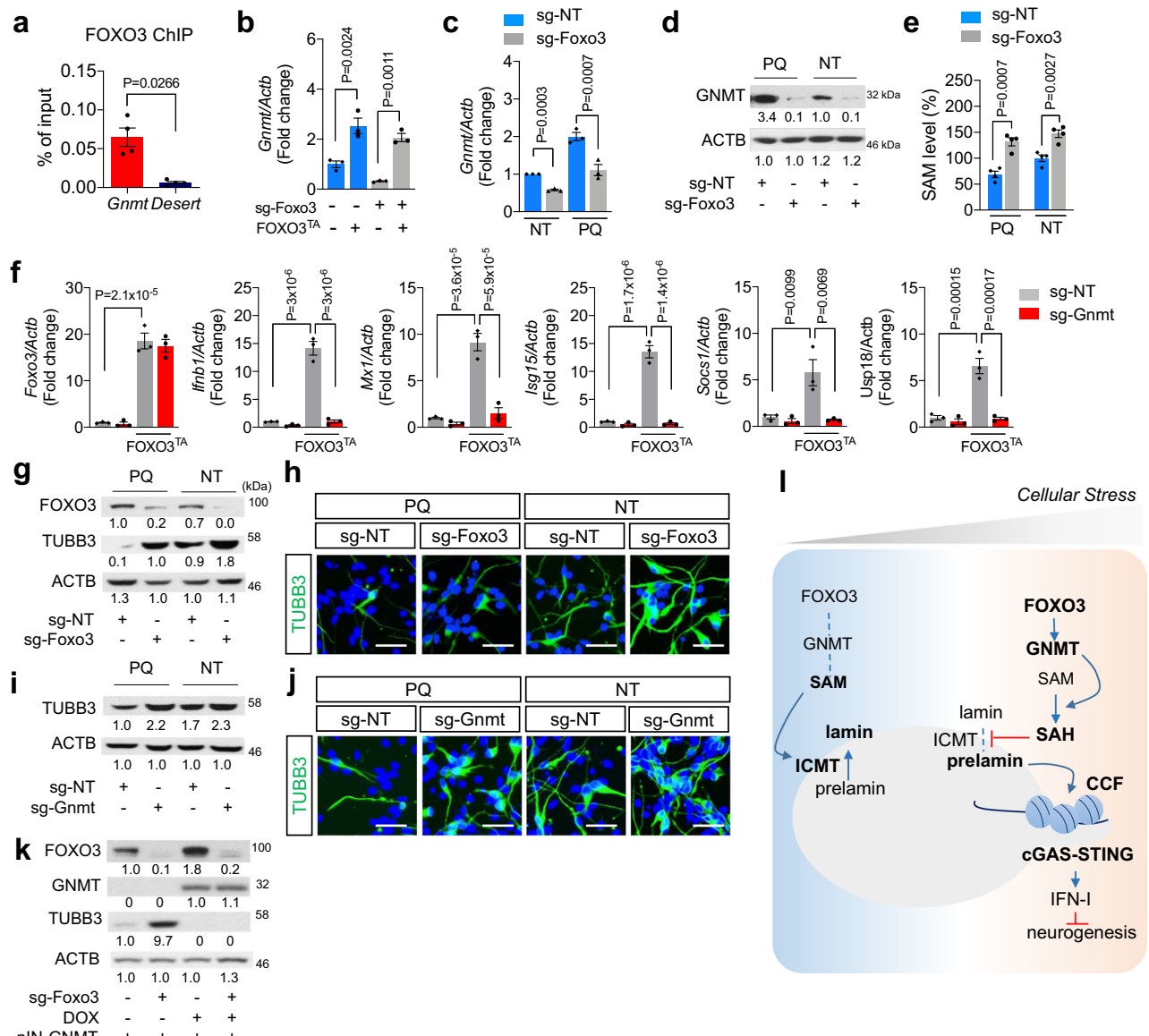

**Fig. 7 Redox stress impacts neurogenic potential of NSPCs through regulating SAM levels. a** FOXO3 ChIP-qPCR analysis at *Gnmt* promoter in comparison to a gene desert region. Mean ± s.e.m. of four independent experiments. qRT-PCR (**b, c**) and WB (**d**) results for GNMT expression in NSPCs treated with PQ for 48 h. Mean ± s.e.m. of three independent experiments. **e** SAM levels in NSPCs expressing sg-NT or sg-Foxo3 treated for 48 h as indicated. Hundred percent of SAM = 50 μM. Mean ± s.e.m. of four independent experiments. **f** qRT-PCR results for Foxo3 and ISGs on either control adenovirus or FOXO3^TA adenovirus-infected NSPC. Mean ± s.e.m. of three independent experiments. Statistical significance was determined by two-sided unpaired *t*-test for **a** and by one-way ANOVA for **b, c, e**, and **f**. WB (**g, i**, and **k**) and IF (**h** and **j**) for TUBB3 expressions of each NSPC line on 3 day of differentiation. Scale bar = 40 μm. Experiments for **d, g, h, i, j**, and **k** were repeated three times independently with similar results and representative images/blots are shown. **k**. Model for the mechanism how cellular stress elicits IFN-I response and inhibits neurogenic differentiation potential of neural stem cells.

SAM-lamin-cGAS/STING-IFN-I signaling cascade as an important physiological stress response program that may protect the nervous system against acute oxidative insults (Fig. 7l).

## Discussion

Alterations of the redox state, as in many brain pathologies, regulate the fate of NSPCs[48]. Our study revealed that cellular stresses including a higher redox potential are translated into IFN-I response via FOXO3-GNMT/SAM-lamin changes (Fig. 7l). In particular, we showed that redox potential controls NSPC function by altering IFN-I response through metabolic regulation of intracellular SAM availability. Mechanistically, our study

uncovered a previously unidentified FOXO3 signaling cascade that functionally connects oxidative stress response with NSPC differentiation through SAM-depletion-induced IFN-I activation. Our findings of redox-dependent neurogenic regulation warrant future studies on the therapeutic rejuvenation of stress-impacted adult NSPCs.

FOXO transcription factors play a central role in a wide range of biological processes, including stress sensing and regulation of stress response[12]. Genetic studies from many organisms have repeatedly demonstrated the conserved insulin/IGF-PI3K-AKT-FOXO cascade as a major regulatory signaling pathway of aging and lifespan. In the central nervous system, expression of FOXO plays not only a key role in preserving neural stem cell pools[9,10],

but also protects neurons against age-related axonal degeneration across species[16–18]. Despite these advances, there still lacks a mechanistic understanding of how oxidative stress affects FOXO activation systematically and whether and how that contributes to the neuroprotective responses. In the current study, we identified FOXO3 oxidation at the evolutionarily conserved Cys31 residue as a regulatory mechanism that modulate redox-dependent FOXO3 nucleo-cytoplasmic shuttling and downstream signaling. Notably, a previous study reported that ROS-induced FOXO4 oxidation at Cys239 promotes its nuclear import by forming a disulfide-dependent protein complex with transportin-1[31]. These findings suggest that redox-regulated nuclear shuttling is a conserved mechanism underlying FOXO-mediated oxidative stress response.

Our data indicate that FOXO3 mediates redox response through regulation of GNMT and downstream SAM catabolism. Enhanced SAM catabolism by GNMT extends the lifespan in *Drosophila*[49]. In the nervous system, GNMT-mediated SAM metabolism is required for the proliferative signaling of NSPC and hippocampal neurogenesis[50]. But the underlying mechanism is unclear. Here we found that treatment of NSPCs with pro-oxidants led to upregulated GNMT expression and reduction of intracellular SAM availability. SAM is a metabolite generated via the one-carbon metabolism and is the main methyl donor in cellular methylation reactions[40,41]. SAM depletion through dietary methionine restriction has been shown to modulate histone methylation and induce stem cell quiescence[43,44,51]. Amongst the plethora of cellular SAM actions, our study found that not only SAM depletion in NSPCs confers a global reduction of H3K4 methylation, but is also sufficient to trigger cGAS/STING signaling and IFN-I response through regulation of nuclear lamin maturation. These findings support FOXO3-GNMT/SAM axis as a stress responsive program that protect tissue homeostasis by orchestrating anti-oxidative function, metabolic rewiring, and gene expression.

Defective lamin processing is known to cause various human pathologies, particularly those related to aging. A truncated lamin A causes a premature aging syndrome of Hutchinson–Gilford progeria. Consistent with our findings, expression of mutant lamin activated IFN-I response[52]. In addition, recent reports suggest that lamins play important roles in both the outside-in and inside-out signaling processes. External mechanical forces trigger changes in nuclear envelope structure and composition, chromatin organization, and gene expression[53]. Likewise, lamin A is stabilized upon external stress to protect the genome[54]. These studies agree with our findings linking lamin and cellular stress response. We found that FOXO3 mediates oxidative stress response through regulation of intracellular SAM availability and nuclear lamin posttranslational modification. Introduction of a C-terminus deletion form of mature LMNA^m blunts ROS-induced activation of cGAS/STING signaling and IFN-I response. These findings suggest a role of nuclear lamin as a signal transducer that mediates oxidative stress response.

Our study contributes to accumulating literature that links cytosolic DNA-sensing cGAS/STING-IFN-I program to physiological and pathological responses to maintain CNS homeostasis. While initially recognized for its critical function in the innate immune response against viral infections, recent studies indicate that cGAS/STING-IFN-I pathway also mediates many other stress responses including signaling from DNA damage and oxidative stress[55–58]. Notably, increased IFN-I response suppresses proliferation of NSPC and reduces their neuronal differentiation under oxidative stress[14,24,59]. In this study, we demonstrated that oxidative stress response activates FOXO3-GNMT/SAM-cGAS/STING-IFN-I signaling cascade and regulates neurogenic potential of cultured NSPCs. Considering increased IFN-I response with declined neurogenesis is an indicative of aging brains[6,15], we propose cGAS/STING-IFN-I response as an intrinsic cellular surveillance system that protects NSPCs against the deleterious consequences of oxidative insults.

Consistent with our findings, previous aging studies reported that lowering systemic SAM levels by dietary restriction of its precursor methionine was effective toward extending lifespan and improving tissue functions in mammals[60]. Engaging FOXO3-GNMT/SAM-lamin-IFN-I response to acute stress conditions is likely to protect organisms against losing long-term regenerative potential. This protective mechanism, nevertheless, may drive stem cell dysfunction by increasing quiescence and decreased differentiation potential at the face of chronic pathological stress stimuli. Altogether, our findings revealed molecular mechanisms that outline how oxidative stress may trigger IFN-I response-mediated cellular protective response and homeostasis under pathophysiological conditions.

## Methods

**NSPC culture and differentiation**. Primary murine NSPCs isolated from sub-ventricular zones (SVZ) and cultured as neurospheres are heterogeneous populations with limited repopulation potential. To avoid passage-dependent drift in NSPC populations, we utilized a neonate-derived immortalized $Ink/Arf^{-/-}$ NSPC culture that maintains the multi-lineage differentiation capability[61]. It contains a mixed population of relatively quiescent neural stem cells, activated NSCs and lineage-committed neuronal precursor cells, as well as oligodendroglial progenitors based on mRNA expression of lineage markers. NSPCs were cultured with N2 media including 20 ng/mL of EGF and bFGF in the presence or absence of 5 mM NAC, 10 μM PQ, or 40 ng/mL Interferon-β. After 2 days, all growth factors and chemicals were removed and changed to N2 media including B27 supplement to induce differentiation. Cells were harvested at indicated time points for analysis. All the sources of materials are listed in Supplementary data 2.

**Generation of viral particles**. To generate lentivirus, $1.5 \times 10^7$ 293T cells in 150 mm tissue culture dishes were transfected with 18 μg of each plasmid DNA along with 4.5 μg of pMD2.G and 9 μg of psPAX2 packaging vectors using poly-ethylenimine. The medium containing lentiviral particles were collected at 48 and 72 h after transfection. The expression of GFP-FOXO3^TA by adenoviral transduction was performed by incubating NSPC culture with pfu of purified adenoviral particles for 16 h. Empty adenoviral particles were used at the same pfu in control cultures. All the sources of plasmid DNAs, materials, viruses, and all oligo sequences are listed in Supplementary data 2.

**Measurement of ROS and redox potential**. Intracellular glutathione redox potential was determined by expressing pLPCX cyto Grx1-roGFP2[62]. Grx1-roGFP2 expressing NSPCs were treated with 10 mM N-ethylmaleimide for 5 min before fixing with 4% paraformaldehyde to prevent further oxidation. Cells from random fields were scanned by Olympus FLUOVIEW laser scanning confocal microscope using excitation at 405 nm/488 nm. Image analysis was performed using ImageJ (version1.52k) software to calculate 405/488 nm ratio.

**Measurement of IFNβ**. The medium was collected following 2 days of treatment. The cells were lysed with laemmli buffer and protein concentrations were determined by BCA assay. IFNβ was measured by using VeriKineTM Mouse IFNβ ELISA kit following the manufacturer's instructions. All the sources of materials are listed in Supplementary data 2.

**SAM assay**. Metabolites were extracted by using cold 80% methanol from 3 million cells overnight at −80 °C. Relative SAM levels were determined by using MLL1 SAMe-Screener Assay kit following the manufacturer's instructions. In brief, all standards and samples were incubated with MLL1 enzyme for 15 min and subsequently SAM-binding site probe for 15 min at room temperature. The levels of free probe for each well were determined by a plate reader (SpectraMax M4) with excitation and emission wavelengths of 575 nm and 620 nm, respectively. All the sources of materials are listed in Supplementary data 2.

**RNA extraction and qRT-PCR analysis**. Total RNAs were extracted from cells by using NucleoSpin RNA kit. Reverse transcription was carried out on 500 ng of total RNA using utilizing RevertAid RT kit. qRT-PCR was performed on cDNA samples using the PowerUp™ SYBR® Green Master Mix on the 7500 Fast Real-time PCR system. All samples were run in duplicate and the mRNA level of each sample was normalized to that of ACTB mRNA. The relative mRNA level was presented as unit values of 2^dCt (=Ct of ACTB-Ct of gene). All the sources of materials and primers are listed in Supplementary data 2.

**RNA-seq and data analysis**. Total RNAs were isolated from NSPCs treated with 5 μM of PQ or 5 mM of NAC for 48 h and subjected to RNA sequencing at the Genomics Resources Core facility of Weill Cornell Medicine. RNA-seq libraries were prepared using the Illumina TruSeq stranded mRNA library preparation kit and sequenced on HiSeq4000 sequencer (Illumina). RNA-seq data were aligned to the mm9 reference genome using TopHat[63], and Cufflinks was used to measure transcript abundances in fragments per kilobase of transcript per million mapped reads[64]. GSEA analysis in this manuscript was generated from the GSEA preranked model[65,66]. The input of GSEA analysis is the gene expression level logFC (fold changes over control).

**Immunofluorescent analysis**. Cells were fixed with 4% paraformaldehyde for 10 min at room temperature followed by permeabilization with 0.2% triton X-100 in PBS. The cells were subjected to immunofluorescence staining with antibodies against β-III tubulin (TUBB3, 1:300), GFAP (1:100), NESTIN (1:300), anti-lamin B1 (1:1,000), 8D1 (1:100), H3K4m3 (1:10,000), or FOXO3 (1:1,00) antibodies overnight at 4 °C. The cells were then washed with cold PBS, and incubated with Alexa 488-labeled or Alexa 594-labeled secondary antibodies at room temperature for 1 h. Images were acquired by fluorescence microscopy with EVOS FL Cell Auto Imaging System. All the sources of antibodies and materials are listed in Supplementary data 2.

**Western blot analysis**. Cells were lysed by laemmli buffer followed by sonication (30 W/5 s/10 cycles). Protein concentration was determined by using Pierce™BCA protein assay kit. Five to 30 μg of proteins were fractionated by SDS-PAGE electrophoresis and transferred to PVDF membrane using a transfer apparatus following manufacturer's instructions. After incubation with 5% skim milk in TBST (10 mM Tris, pH 8.0, 150 mM NaCl, 0.5% Tween 20) for 1 h, the membrane was incubated with antibodies against β-actin (ACTB, 1:10,000), βIII Tubulin (TUBB3, 1:5,000), H3K4m3 (1:10,000), Histone H3 (1:10,000), GNMT (1:500), FOXO3 (1:1,000), p-FOXO1/3a (T24/T32, 1:1,000), p-TBK1 (S172, 1:1,000), p-STAT1 (Y701, 1:1,000), p-Akt (S473, 1:2,000), Akt (1:5,000), p-PRAS40 (T246, 1:1,000), ICMT (1:1,000), lamin B1 (8D1, 1:500), lamin B1 (1:5,000), or IFNAR2 (1:1,000) overnight at 4 °C. Membranes were washed three times with TBST for 30 min and then incubated with HRP conjugated anti-mouse or anti-rabbit diluted in 3% skim milk for 1 h. Blots were developed with the SuperSignal™ West Pico Chemiluminescent substrate according to the manufacturer's protocols. All the sources of materials and antibodies are listed under Supplementary data 2.

**Immunoprecipitation for detecting Cys-sulfenylation**. Cells were pretreated with 2 μM of dimedone for 1 h and then, treated with 100 μM of PQ or 200 μM of hydrogen peroxide for 6 h or 2 h, respectively. Cell were lysed using 1% TNT buffer (135 mM NaCl, 20 mM Tris-HCl, 1 mM EDTA, and 1% Triton X-100) for 30 min on ice. After centrifugation to remove the debris, 1 mg of protein was incubated with 10 μL of Anti-DYKDDDDK Magnetic Agarose overnight at 4 °C. Beads were washed three times with 1% TNT buffer and the proteins were eluted with 2× Laemmli buffer. The protein interaction was determined by western blot. All the sources of materials and antibodies are listed in Supplementary data 2.

**Chromatin immunoprecipitation (ChIP)**. ChIP analysis was performed following the previous report[67]. In brief, $3 \times 10^7$ cells were crosslinked for 5 min with 1% paraformaldehyde and quenched with 125 mM glycine for 5 min at room temperature. After nuclei isolation, the chromatin was sheared in shearing buffer (50 mM Tris-HCl, 10 mM EDTA, and 0.1% SDS) using the Covaris M220 focused-ultrasonicator according to the manufacturer's instructions. Immunoprecipitation was performed with 10 μg of anti-FOXO3 overnight at 4 °C. Thirty microliters of pre-cleared Dynabeads® Protein G was added and incubated for 3 h at 4 °C. The beads were washed with RIPA buffer (including LiCl) and eluted with elution buffer (50 mM Tris-HCl, 10 mM EDTA, and 1% SDS). After RNase and Proteinase K treatment, eluted DNA was reverse-crosslinked by 65 °C incubation overnight. DNA was extracted using NucleoSpin Gel and PCR clean-up DNA extraction kit and size-selection was carried out to obtain <400 bp size DNA fragments using SPRIselect Reagent. qRT-PCR was performed using specific primers. All the sources of materials are listed under Supplementary data 2.

**Metabolomics analysis**. Ten million cells were homogenized in cold 80% methanol using homogenizer. Metabolites were extracted over 3 h at −80 °C. Samples were then centrifuged at 4 °C for 15 min at 14,000 rpm. The supernatants were extracted and normalized based on tissue weight. Targeted LC/MS analyses were performed on a Q Exactive Orbitrap mass spectrometer coupled to a Vanquish UHPLC system. The Q Exactive operated in polarity-switching mode. A Sequant ZIC-HILIC column (2.1 mm i.d. × 150 mm) was used for separation of metabolites. Flow rate was set at 150 μL/min. Buffers consisted of 100% acetonitrile for mobile A, and 0.1% $NH_4OH$/20 mM $CH_3COONH_4$ in water for mobile B. Gradient ran from 85% to 30% A in 20 min followed by a wash with 30% A and re-equilibration at 85% A. Metabolites were identified on the basis of exact mass within 5 ppm and standard retention times. Relative metabolite quantitation was performed based on peak area for each metabolite.

**Transient middle cerebral artery occlusion (tMCAO) model**. Mice were group housed (up to 5 per cage) in individually ventilated cages with ad libitum access to food and acidified water (pH 2.5 to 2.8) in a temperature (22.2 ± 0.5 °C) and humidity (30–70%) controlled facility with 12:12-h light:dark cycle. The animal care and use program is accredited AAALAC. All animal experiments were approved by the Weill Cornell Institutional Animal Care and Use Committee. Eight- to ten-week-old mice (C57BL/6J) were subjected to the intraluminal suture-induced tMCAO model[68,69]. During the procedure, mice were deeply anesthetized with Isoflurane and their body temperature was maintained at 37 °C by a self-regulating heating pad. The intraluminal suture was used for the middle cerebral artery occlusion for 60 min followed by reperfusion. The success of the surgery was confirmed by measuring the blood flow in the territory of middle cerebral artery with laser speckle contrast analysis (PeriCam PSI HR). For sham surgery, animals were anesthetized and subjected to the same surgical procedures except insertion of the suture in the MCA. Brains were harvested for histological sectioning and protein/RNA extraction 3, 6, and 24 h after tMCAO.

**Statistical analysis**. We determined experimental sample sizes on the basis of preliminary data. All results are expressed as mean ± s.e.m. GraphPad Prism software (version 7.0e) was used for all statistical analysis. Normal distribution of the sample sets was determined before applying unpaired Student's two-tailed $t$ test for two group comparisons. One-way ANOVA was used to assess the differences between multiple groups. The mean values of each group were compared by the Bonferroni's post-hoc procedure. Differences were considered significant when $P < 0.05$.

**Reporting summary**. Further information on research design is available in the Nature Research Reporting Summary linked to this article.

## Data availability
The data that support the findings of this study are available within the article and its Supplementary Information files. All the uncropped western blots and raw data are provided as a Source data file. RNA-seq data can be found on Gene Expression Omnibus (GEO) database with accession number GSE146243. Source data are provided with this paper.

## Code availability
For specific requests, please contact the corresponding author.

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

## Acknowledgements

Authors thank Dr. Loren Fong (UCLA) for kindly providing prelamin antibodies and Drs. Claire Vanpouille-Box and Silvia Formenti for discussion. We also thank Drs. Zhe Cheng, Guoan Zhang at the Proteomics and Metabolomics and Tuo Zhang at the Genomics Resources Core Facility of The Weill Cornell Medicine for analysis. This work was supported by the Irma T. Hirschl Award, the Anna Maria and the Kellen foundation, National Institutes of Health Grant AG048284 (to J.P.), and NS114561 (to T.S.)

## Author contributions

Conception and design: I.H., H.Z., J.P. Development of methodology: I.H., H.Z. Acquisition of data: I.H., H.U., F.L., J.P. Analysis and interpretation of data (e.g., statistical analysis, biostatistics, computational analysis): I.H., H.U., Z.D., J.W.L., J.P.

Writing, review, and/or revision of the manuscript: I.H., H.Z., J.P. Administrative, technical, conceptual or material support (i.e., reporting or organizing data, constructing databases): I.H., J.W.L., L.C.C., T.S., H.Z., J.P. Study supervision: J.W.L., T.S., J.P.

## Competing interests

L.C.C. is a founder and member of the BOD of Agios Pharmaceuticals and is a founder and receives research support from Petra Pharmaceuticals. These companies are developing novel therapies for cancer. Other authors have declared that no competing interests exists.
