## [Peer Review File · Nature Communications]

REVIEWER COMMENTS

Reviewer #1 (Remarks to the Author):

The manuscript by Hwang and colleagues identifies a novel FOXO3-mediated oxidative stress response in NSPCs and differentiating neurons through the induction of glycine-N-methyltransferase, altered lamin processing, and activation of CGAS/STING/IFN-1 signaling. Specifically, the authors observe that paraquat (PQ) treatment induces an IFN-1 response and reduces NSPCs ability to differentiate into neurons. Oxidative stress increases levels of FOXO3 and its activation in NSPCs. The authors observe that FOXO3 is necessary for the IFN-I response in PQ treated cells and that constitutive activation of FOXO3 induced a similar IFN-I activation. Interestingly, PQ treatment results in disrupted lamin processing and leakage of nuclear material, which the investigators link to a depletion of SAM and disrupted lamin processing. Mechanistically, the reduction in SAM is caused by FOXO-induced glycine-N-methyltransferase.

Overall, this study is exciting and novel. Previous studies, including work from this group, showed the importance of FOXO3 in maintaining the stem cell pools in the brain. In this study the authors make the novel observation that oxidation at Cystine 31 on FOXO3 is correlated with a reduction in phosphorylation of Threonine 32, suggesting a new mechanism by which oxidative stress may activate FOXO3. Moreover, the identification of GNMT as a target, and the subsequent links to lamin processing and cGAS/STING/IFN-1 are new findings that will be of general interest. Overall, the proposed mechanisms regulation is supported by the evidence presented in the manuscript. There are however some points in this study that should be addressed before publication.

Major points:

One question raised by the study is regarding the fate and status of the NSPCs under oxidative stress. The authors stated in lines 103-105 that "compared to the mock-treated control NSPCs, we found that NSPCs under PQ but not NAC treatment, exhibited marked reduction in production of TUBB3 or doublecortin (DCX)-positive newly born neurons when induced to differentiate" but it does not appear there is a strong change in DCX. This could be indicative of a difference in maturation of new neurons instead of commitment of NSPCs to differentiation. Also when neurogenesis is reduced, what happens to the rest of the lineage? Is differentiation to oligodendrocytes also impaired? Do the cells express markers of astrocytes (s100b?) or retain markers of NSPCs (nestin?)? In general, it would be useful to have quantification of the immunocytochemistry experiments, which would clarify the magnitude of the neurogenesis defects under different conditions.

Fig 1e. Information on what is presented in the heat map is missing. Are these fold changes? There should be a scale indicating the values. In addition, the entire set of GSEA results should be presented in a supplemental table. It also appears that the RNA-seq experiment was performed under proliferating conditions. Is this correct? If so, it would be important to assess how the dividing NSPC pool is affected since the INF-1 response is induced (related to the above point). Or, is this effect specific to the differentiating cells?

The authors provided evidence showing that under oxidative stress conditions NSPCs had elevated levels of IFN β secretion, and that this phenotype was suppressed upon knockdown of FOXO3. However, there is no experiment showing that the IFN β secretion was similarly increased under conditions where FOXO3 was constitutively activated. Does FOXO3 activation affect IFN β secretion?

Figure 3C shows increased FOXO3 levels in the NSPCs with PQ treatment but this seems to be inconsistent with the data shown in 3F, where the levels of FOXO3 appeared to remain constant between the control cells and those treated with PQ. Could the authors provide information on how

loading was controlled in this experiment and why the result appears to be different from 3C as the total FOXO3 levels does not appear to change with PQ?

The study includes an immunoprecipitation experiment, where the authors showed that there was increased FOXO3 sulfenylation in ROS treated NSPCs, when compared to the controls. In this experiment hydrogen peroxide was used to induce oxidation while in other experiments in this study were performed with PQ. Is the same effect oxidizing effect observed with PQ?

Minor points:

Line 155. This sentence is referring to Fig 3f? This should be included in the text.

The study includes a ChIP-qPCR experiment however it is not indicated in the methods section of the manuscript the antibody that was used for this experiment. All the antibodies used for this and other experiments should be included.

Reviewer #2 (Remarks to the Author):

The authors report a signalling pathway that prevents differentiation and depletion of NSCPs under oxidative conditions. This is a novel finding and is supported by their data. They show that treatment with paraquat to induce oxidative damage, suppresses the formation of new neurons from NSCPs and this suppression appears to depend on IFN-1. They then show that FOXO3 is involved in the IFN-1 response. They link FOXO3 activation to increased GNMT and a depletion of SAM required for lamin methylation. Improper lamin maturation results in activation of the DNA sensing cGAS/STING pathway.

Their data are convincing, however a weakness of the study is that most of the data are from cultured NSCPs. However they do confirm some results in aging postmortem brains. The postmortem interval (PMI) and sex for these samples should be given. Also, why did they analyze cerebellum?

The conclusions would be strengthened by adding some in vivo data showing at least some components of the proposed pathway (reduced FOXO3 in the nucleus or an increase in sulfenylation, increased GNMT, changes in ISGs) in NSCPs in aged mice or in mice exposed to oxidizing conditions.

What isn't clear, is why paraquat was chosen to induce redox dysregulation. Also, the timing of paraquat treatment of NSCPs was 48 hours for experiments to measure gene expression changes. It's difficult to know if this paraquat treatment and subsequent gene expression changes are relevant to the in vivo condition. A dose response should be done. Are gene expression changes after 48 hours consistent with chronic redox changes that occur during aging in vivo?

Another weakness is that the link between the IFN-1 pathway and regulation of NSCP differentiation is tenuous. The depletion of SAM by paraquat may be extreme and may be causing artifacts in NSCPs. They show that FOXO3 activates GNMT expression and activity that depletes SAM. The decrease in SAM would have many non-specific effects on many different methylation reactions however, this isn't discussed. They do measure a decrease in H3K4me3. Have they looked at any other methyltransferases? Could the inhibition of differentiation of NSCPs in to new neurons be a result of inhibition of DNMTs and changes in DNA methylation? Possibly the IFN-1 pathway is a separate response that isn't directly related to neuronal differentiation? This possibility should be addressed.

Reviewer #3 (Remarks to the Author):

Hwang et al report their findings of a molecular mechanism involving FOXO3, nuclear lamins, SAM-methyl donor and innate immunity, which they propose to form a protective response in neural stem cells upon acute oxidative stress, to maintain stemness. In more detail, the authors utilize paraquat, a ROS-generating toxin, and report activation of FOXO3 and upregulation of GNMT to deplete SAM, which they propose to disrupt the nuclear lamin maturation. Moreover, they suggest that compromised nuclear envelope integrity further activates innate-immunity through cGAS/STING-IFN-1 signaling cascade to regulate neurogenesis under oxidative stress conditions. The proposal is intriguing and the pathway component links are partially but not completely established. Despite this, the findings of GNMT-SAM-FOXO3 and lamin links to innate immunity induction are interesting. Overall, the manuscript is well-written and experiments are mostly of good technical quality with data supporting the conclusions. However, quite a few points are essential to address:

1. The authors study cultured neural stem cells. However, whether the reported mechanistic findings are specific to NSCs or generally relevant for other cells has not been studied. Some primary non-stem cell lines should be tested, such as MEFs.
2. Paraquat is the ROS-inducer and main insult used in the paper. However, it is a toxin, and acts through many mechanisms. Still no consensus exists, which are the most relevant effects in different cell types. The increase of oxygen radicals, linked to impaired iron metabolism, lipid peroxidation, and ultimately ferroptosis all contribute to cell dysfunction and death. What is the amount of cell death with the used dose of paraquat in their NSCs?
3. To use paraquat as a ROS-generator and conclude that the effects of this severe toxin are physiologically relevant is quite far fetched. In Figure 3E they use H₂O₂ for some reason, but elsewhere paraquat. H₂O₂ would be an excellent and more physiological tool for ROS effect to replicate some key findings.
4. Neural stem cells can be identified by IHC in vivo, in subventricular zone of mice. An in vivo paraquat-exposure and analysis of specific readouts (FOXO3 in nucleus; nuclear membrane abnormality in stem cells, IHC of IF-β linked genes or GNMT) would point to in vivo relevance of the findings and indicate specificity / non-specificity to stem cells and strengthen the paper considerably.
5. In figure 1, the authors suggest that type I interferon response is the most enriched pathway upon PQ treatment accompanied by reduced production of TUBB3 and DCX during neuronal differentiation (Figure 1D and 1E). Figure 1D is not clear, and inadequately explained in the text and legend. Please revise. Furthermore, increased TUBB3 and DCX were detected upon NAC treatment. Do the authors observe reduced expression of IFN I signaling players upon NAC treatment with or without paraquat?
6. A complete set of data of the gene expression profiling of differentially treated NSPCs should be added to the supplement.
7. In Figure 1G and 1F, significant amount of IFNβ was shown to be released from the cells upon PQ treatment and IFNβ treatment could inhibit the production of TUBB3 during neuronal differentiation. IFNAR is the ubiquitous receptor for type I interferon cytokines. Does depletion of the receptor on NSPCs prior to PQ treatment also attenuate activation of type I interferon signalling and

neuronal differentiation? This finding would further support the specificity of these defects to type I interferon cytokines. A negative control to assess for the activation of pro-inflammatory cytokines like IL-1beta or IL-6 should also be included.

8. In Figure 2, the authors show that FOXO3 depletion attenuated PQ-induced IFN I signaling. Mice deficient of FOXO family members including FOXO1, FOXO3 and FOXO4 have excessive oxidative stress which in turn decreases HSC number and their long-term regenerative potential (Tothowa et al., 2007, Cell). Do FOXO1 and FOXO4 depletion also contribute to this immune signaling restoration, or is the effect in neural stem cells FOXO3 specific ?

9. In Figure 4, the authors observed abnormal nuclei and cGAS-GFP foci upon PQ treatment indicating compromised nuclear envelope integrity, and showed increased cGAS foci upon expression of GNMT in Figure 6K. The authors claim that nuclear DNA leakage induces activation of cGAS/STING pathway following GNMT activation and insufficient lamin maturation. Their DNA-leakage conclusion is only based on nuclear morphology and detection of cGAS/STING foci in the cells. A more direct evidence of DNA leakage following oxidative stress elevation is needed, for instance, direct immunofluorescence analysis visualizing DNA to show increased cytosolic DNA (present outside of nuclei), upon treatment with PQ or GNMT overexpression. Restoration of interferon signaling following depletion of cGAS or STING in these cells prior to oxidative stress elevation would further support the role of cGAS-STING pathway in NPSCs.

10. The experiments using FOXO3-C31A mutant are excellent, providing clear views to the mechanisms. Could the C31A mutant of FOXO3 induce IFN-beta upon PQ?

11. The finding of the link of low SAM, activation of cGAS/sting and induction of IFbeta is solid and interesting. Whether all their findings are mechanistically linked, or whether several pathways may lead to IFNbeta induction, is not clear. However, paraquat is a toxin, with several deleterious effects. Whether the pathways reported actually are a physiological mechanism that regulates neurogenic potential is not verified by the paper. The discussion of relevance to aging and regulation of NSC neurogenesis, without any in vivo data, is too far fetched and should be omitted.

12. A quantification of western blotting results and kDa molecular mass for proteins identified should be provided. As many westerns (e.g. 2b) are of suboptimal quality, the whole gels should be included in supplementary data.

Reviewer #4 (Remarks to the Author):

The manuscript by Hwang et al. proposes a role for FOXO3-mediated regulation of lamin post-translational modification (carboxymethylation) on the response to cellular stress that activates the cGAS-STING pathway and the downstream interferon (IFN) response. In addition, they propose that activation of this cascade suppresses neurogenic differentiation of neural stem/progenitor cells (NSPCs). The study touches a subject that is understudied and that has received high attention in recent years: the connection between cellular stress, the IFN response, and stem cell differentiation potential. Overall, the study is significant for the field, describing new molecular mechanisms linking oxidative stress to activation of inflammatory cascades such as the IFN response. It also highlights the importance of proper post-translational processing of lamins not only for nuclear function but also for processes that take place at the cytoplasm, such as cGAS/STING/IFN pathway activation, and that are key for cellular homeostasis. The findings are innovative and of potential high impact across different fields.

Strengths:

- The manuscript is well-written, the experimental design is for the most part appropriate to test the different hypotheses, and the conclusions are in accordance with the results. In addition, the proper controls are included, with some exceptions noted below.
- Strong evidence is provided for an impact of oxidative stress inhibiting NSPCs' differentiation, as well as RNAseq evidence for activation of an IFN response.
- Convincing data also support the notion that translocation of FOXO3 into the nucleus and activation of downstream effectors mediates IFN response activation, with Cys31 sulfenylation and Thr32 phosphorylation being important.
- Moreover, the authors make a clear-cut connection with lamins post-translational processing of their C-terminal -CAAX motif. In particular, the C-terminus of lamins seems to play a role in the activation of IFN response by PQ. Proper processing of lamins is critical to maintain IFN pathway under control, and hindering processing activates the pathway. A special role is shown for carboxymethylation of terminal Cys in lamins, which is inhibited upon PQ due to reduction in SAM levels triggered by upregulation of GNMT. These findings lead the authors to conclude that GNMT upregulation and SAM depletion elicit deficiencies in lamins carboxymethylation that cause nuclear fragility and activation of cGAS/STING and downstream IFN signaling. The importance of lamins in this process is clearly demonstrated by expression of a mutant lamin A protein that lacks the C-terminus. If lamins cannot be methylated, the problem of overexpression of GNMT activated the IFN response is solved.
- The authors elegantly close the circle by showing a connection between FOXO3 activation and GNMT induction under oxidative stress conditions, which plays a role triggering the IFN response and downstream effectors. In addition, they demonstrate that FOXO3 and GNMT upregulation upon PQ treatment hinder the differentiation of NSPCs.

Weaknesses:

1. Experiments of oxidative stress were performed only with paraquat. Other ways to induce oxidative stress could be tested to make sure that this is not a paraquat-specific effect. Along the same lines, the RNAseq on NAC treated samples is missing. Showing that NAC does not activate the IFN response would reinforce the idea that the trigger of this response is oxidative stress. In the RNAseq analysis, did they find upregulation of FOXO targets upon PQ treatment?
2. Depletion of FOXO3 reduces the IFN response in response to PQ but also basal IFN response, suggesting that FOXO3 might regulate the IFN response in all contexts, independent of oxidative stress. This is in fact the case when the authors express the active form of FOXO3. Thus, the conclusion by the authors that "FOXO3 is directly responsible for oxidation stress-induced IFN-I activation" is not totally accurate. FOXO3 seems to activate the IFN response independent of oxidative stress. A question that arises from these data is whether FOXO3 would be a key regulator of the IFN response due to DNA damage, mitochondrial dysfunction, or other stresses. Additional experimental data that addresses whether this is a general mechanism or specific for oxidative stress would improve the quality/significance of the manuscript.
3. The cGAS/STING/IFN pathway is considered upstream of the IFN response, however, while the IFN response is monitored extensively, less is done to test the activation of cGAS/STING pathway. To make the claim that cGAS/STING pathway is upstream of all the phenotypes, inhibition of this pathway should be performed.

Other comments:

1. Why do they use H₂O₂ only in experiment in Fig 3e? Is there FOXO3 sulfenylation upon PQ treatment?
2. In Figure 3h and 3i, the control of PQ treatment in FOXO3 KO cells is missing.
3. In Figures 7g-k it is not clear what High and Low means. Are they different concentrations of PQ?

Point-by-point Response to the Reviewers' Comments

We thank the Reviewers for taking the time to understand our work and the implication it has for the cellular stress induced type-I IFN response on neural stem cell biology. In the revised manuscript, we addressed all Reviewers' suggestions fully. It improved the message as a result.

Reviewer #1 (Remarks to the Author):

Overall, this study is exciting and novel. Previous studies, including work from this group, showed the importance of FOXO3 in maintaining the stem cell pools in the brain. In this study the authors make the novel observation that oxidation at Cystine 31 on FOXO3 is correlated with a reduction in phosphorylation of Threonine 32, suggesting a new mechanism by which oxidative stress may activate FOXO3. Moreover, the identification of GNMT as a target, and the subsequent links to lamin processing and cGAS/STING/IFN-1 are new findings that will be of general interest. Overall, the proposed mechanisms regulation is supported by the evidence presented in the manuscript. There are however some points in this study that should be addressed before publication.

We thank the reviewer for the enthusiasm and supportive comments on our study.

Major points:

One question raised by the study is regarding the fate and status of the NSPCs under oxidative stress. The authors stated in lines 103-105 that "compared to the mock-treated control NSPCs, we found that NSPCs under PQ but not NAC treatment, exhibited marked reduction in production of TUBB3 or doublecortin (DCX)-positive newly born neurons when induced to differentiate" but it does not appear there is a strong change in DCX. This could be indicative of a difference in maturation of new neurons instead of commitment of NSPCs to differentiation.

The discrepancy between the protein levels of TUBB3 and DCX is largely due to the different antibody sensitivity. The anti-TUBB3 (Tuj1) antibody is particularly strong. To avoid unnecessary confusion, we have removed the DCX immunoblot result from the revised manuscript. Also following the suggestion, we now have included the quantitation of TUBB3 cell

percentage as Fig. 1c (right panel). It is noteworthy that the staining was performed on NSPC cells on day 3 following the incipience of differentiation induction. We did not detect any NeuN+ mature neurons at that differentiation stage. Since the Tuj1 antibody used in this study reacts with TUBB3, which is a pan-neuronal marker, we concluded that oxidative stress mostly affects NSPC fate commitment during differentiation, although we cannot rule out the possibility that ROS may additionally influence the process of neuronal maturation.

Also when neurogenesis is reduced, what happens to the rest of the lineage? Is differentiation to oligodendrocytes also impaired? Do the cells express markers of astrocytes (s100b?) or retain markers of NSPCs (nestin?)? In general, it would be useful to have quantification of the immunocytochemistry experiments, which would clarify the magnitude of the neurogenesis defects under different conditions.

Besides neurogenesis, we found that astrocytic differentiation (GFAP+) was also reduced following PQ treatment (Reviewer Figure 1). There was no significant difference in Nestin+ population between the treated and control groups following 3 days of differentiation induction. Interestingly, there was also a clear increase of PDGFRa+ population following PQ treated group. We are currently studying this phenotype.

Fig 1e. Information on what is presented in the heat map is missing. Are these fold changes? There should be a scale indicating the values. In addition, the entire set of GSEA results should be presented in a supplemental table. It also appears that the RNA-seq experiment was performed under proliferating conditions. Is this correct? If so, it would be important to assess how the dividing NSPC pool is affected since the INF-1 response is induced (related to the above point). Or, is this effect specific to the differentiating cells?

We are sorry for the negligence. The heat map in Fig. 1e is based on the Z-score values. The Z-score scale and the description are now added to Fig. 1e. Also following the suggestion, the entire set of GSEA results is now included as Supplementary Table 1. As to RNA-seq, the samples were indeed prepared from proliferating NSPCs. Following the reviewer's suggestion, we have now characterized the cycling property of NSPCs under PQ or IFN β treatment. The results showed that both treatments caused a mild reduction of cell proliferation, as judged by the percentage of S-phase fraction (Reviewer Figure 2). Moreover, immunoblot analysis of PQ or H₂O₂-treated NSPCs against cleaved caspase 3 suggests that the treatment under the dosage used in our study does not cause significant apoptosis. This result is included as Supplementary Fig. 1a.

Reviewer Figure 2. Lack of substantial effect on cycling properties of NSPCs. NSPCs were treated with PQ (10 μ M, 48 h) or IFN β (10 ng/ml 48 h) and subjected to propidium iodide staining and flow cytometry analysis.

The authors provided evidence showing that under oxidative stress conditions NSPCs had elevated levels of IFN β secretion, and that this phenotype was suppressed upon knockdown of FOXO3. However, there is no experiment showing that the IFN β secretion was similarly increased under conditions where FOXO3 was constitutively activated. Does FOXO3 activation affect IFN β secretion?

Thanks for the suggestion. We have now determined the impact of FOXO3-TA expression on IFN β secretion. This result is included as Fig. 2g.

Figure 3C shows increased FOXO3 levels in the NSPCs with PQ treatment but this seems to be inconsistent with the data shown in 3F, where the levels of FOXO3 appeared to remain constant between the control cells and those treated with PQ. Could the authors provide information on how loading was controlled in this experiment and why the result appears to be different from 3C as the total FOXO3 levels does not appear to change with PQ?

Sorry for the confusion. Fig.3c showed endogenous FOXO expression following PQ treatment, whereas Fig.3f was to compare exogenously transduced wild-type and C31A mutants. FOXO3 is known to transcriptionally activate its own promoter². Given that ROS treatment stimulates FOXO3 nuclear retention and activation, the increased endogenous FOXO3 expression observed in PQ-treated cells of Fig.3c likely reflects this self-reinforcing positive feedback loop.

The study includes an immunoprecipitation experiment, where the authors showed that there was increased FOXO3 sulfenylation in ROS treated NSPCs, when compared to the controls. In this experiment hydrogen peroxide was used to induce oxidation while in other experiments in this study were performed with PQ. Is the same effect oxidizing effect observed with PQ?

The reason that we initially chose H₂O₂ instead of PQ was simply due to a technical consideration. This particular experiment requires co-treatment of oxidative stressor with dimedone. Dimedone is necessary for the accumulation and detection of sulfenylated protein under the oxidative stress condition. Since PQ treatment induces ROS through cyclic depletion of the anti-oxidative buffering capacity and generally takes longer time to establish the redox potential, we were afraid that its co-treatment with dimedone might generate high cellular

toxicity. By comparison, H₂O₂ is fast acting and requires much shorter time (<1 h) to build high redox potential and therefore highly amenable to an acute combination. But nevertheless, we have now performed a new experiment by using PQ treatment. The data indicate that PQ treatment of NSCPs, as in H₂O₂ treatment, induced FOXO3 sulfenylation in a dose-dependent manner. This new piece of result is now included as Supplementary Fig. 3b.

Minor points:

Line 155. This sentence is referring to Fig 3f? This should be included in the text.

Sorry for the confusion. It was intended to refer to the two previous studies (ref 32, 33). We have now revised the text to increase the clarity.

The study includes a CHIP-qPCR experiment however it is not indicated in the methods section of the manuscript the antibody that was used for this experiment. All the antibodies used for this and other experiments should be included.

We have now included the antibody information under Supplementary table 2.

Reviewer #2 (Remarks to the Author):

The authors report a signalling pathway that prevents differentiation and depletion of NSCPs under oxidative conditions. This is a novel finding and is supported by their data. They show that treatment with paraquat to induce oxidative damage, suppresses the formation of new neurons from NSCPs and this suppression appears to depend on IFN-1. They then show that FOXO3 is involved in the IFN-1 response. They link FOXO3 activation to increased GNMT and a depletion of SAM required for lamin methylation. Improper lamin maturation results in activation of the DNA sensing cGAS/STING pathway.

We appreciate the reviewer's positive comments and encouragement.

Their data are convincing, however a weakness of the study is that most of the data are from cultured NSCPs. However they do confirm some results in aging postmortem brains. The postmortem interval (PMI) and sex for these samples should be given. Also, why did they analyze cerebellum?

In the revised Supplementary Fig. 9a, we have now marked each data point to show the sex. The samples were procured, and PMI was determined by NIH Brain Biobank. The detailed sample information including sex and PMI is now included as Supplementary Fig. 9b. The choice of cerebellum was strictly based on the sample availability. Although not totally equal, our previous study of old/aged mice noticed an increased type-I IFN signature in other regions of animal brains³.

The conclusions would be strengthened by adding some in vivo data showing at least some components of the proposed pathway (reduced FOXO3 in the nucleus or an increase in sulfenylation, increased GNMT, changes in ISGs) in NSCPs in aged mice or in mice exposed to oxidizing conditions.

To follow the reviewer's suggestion and examine our findings under an in vivo context, we have now performed in vivo experiments using transient middle cerebral artery occlusion (tMCAO)-reperfusion procedure. The tMCAO-reperfusion is a well-characterized brain oxidative stress model^{4,5}. Our immunofluorescence analysis of Nestin+ progenitor cells at SVZ regions of the

animals showed a time-dependent increase of nuclear FOXO3 following stress challenge that was accompanied by loss of 8D1-positive mature lamin expression. Immunoblot and qPCR analysis of the microdissected SVZ tissues of the challenged animals further revealed the increased expression of FOXO3, GNMT as well as other FOXO3 downstream and ISG genes as compared to the control animals. These new pieces of results are now included as Supplementary Fig. 8a-f. We hope that this effort may help to alleviate the reviewer's concern.

What isn't clear, is why paraquat was chosen to induce redox dysregulation. Also, the timing of paraquat treatment of NSCPs was 48 hours for experiments to measure gene expression changes. It's difficult to know if this paraquat treatment and subsequent gene expression changes are relevant to the in vivo condition. A dose response should be done. Are gene expression changes after 48 hours consistent with chronic redox changes that occur during aging in vivo?

We appreciate the opportunity to explain the rationale for our choice of PQ instead of H₂O₂ as the oxidative stressor in many of our experiments. PQ acts through cyclic depletion of anti-oxidative buffering capacity and therefore its effect on cellular ROS is indirect but durable. By contrast, more commonly used oxidative stressor H₂O₂ is direct but rather transient in its effect. This was demonstrated in a previous study by measuring endogenous ROS levels following PQ or H₂O₂ treatment (Reviewer Fig. 3) ¹.

[Redacted]

Reviewer Figure 3. #Figure 1C-D adopted from *Wiemer et al.* (2014) ¹. Quantitative measurement of H₂O₂ release by H₂O₂ HyPer probe in p anserina. H₂O₂ (left) or PQ (right) were added at arrow indicated time points. Note rapid and transient effect by H₂O₂ and gradually increasing effect by PQ following a bolus treatment.

Since in vitro neuronal differentiation takes place over several days, we chose PQ as the main oxidative stressor based on its durable effect on maintaining experimental intracellular redox potential. However, it was never our intention to claim that our in vitro system and their derived gene expression changes can totally recapitulate the in vivo or aging conditions. Nevertheless, our previous study did reveal increase of both FOXO and type-I IFN signatures in old/aged animal brains ³. We are also gratified to see that major findings from our in vitro system could be validated by the new tMCAO-reperfusion in vivo experiments.

Another weakness is that the link between the IFN-1 pathway and regulation of NSCP differentiation is tenuous. The depletion of SAM by paraquat may be extreme and may be causing artifacts in NSCPs.

We agree with the reviewer that paraquat treatment may produce undocumented effect. But our finding that activated FOXO3 (TA mutant) also triggers SAM depletion through upregulating GNMT suggests that this ROS-mediated process is not specifically caused by paraquat treatment. Also, our finding that addition of IFN β alone was sufficient to suppress neurogenic differentiation of NSPCs (Fig. 1h) supports a direct link between IFN-1 signaling and NSPC differentiation regulation.

They show that FOXO3 activates GNMT expression and activity that depletes SAM. The decrease in SAM would have many non-specific effects on many different methylation reactions however, this isn't discussed.

We also agree with the reviewer that cellular SAM action is complex, and our study only touched one of many physiological roles of SAM. We have modified the text following the reviewer's suggestion.

They do measure a decrease in H3K4me3. Have they looked at any other methyltransferases? Could the inhibition of differentiation of NSCPs into new neurons be a result of inhibition of DNMTs and changes in DNA methylation? Possibly the IFN-1 pathway is a separate response that isn't directly related to neuronal differentiation? This possibility should be addressed.

The reviewer asked a very important point. Initially we also hypothesized methylation-dependent epigenetic alteration as the mechanism that underlies SAM depletion-caused neurogenic differentiation change. To test that hypothesis, we isolated FOXO3 wild type (WT) and knockout (KO) NSPCs from mouse brain and performed Enhanced Reduced Representation Bisulfite Sequencing (ERRBS)⁶. This allowed us to determine changes of global cytosine methylation levels (% mC) in SAM elevated FOXO3 KO NSPCs. Our results suggested that there was no remarkable global methylation differences between FOXO3 WT and KO NSPCs (Reviewer Fig. 4 left). Consistently, principle component analysis and unsupervised clustering also did not segregate WT from FOXO3 KO NSPCs (Reviewer Fig. 4 right).

Since there were no remarkable global methylation differences between FOXO3 WT and KO NSPCs, we next tested whether SAM depletion regulates histone methylation. Among many methyltransferases, MLL1 was shown to be most sensitive to the lowered SAM levels due to its high Km value^{7,8}. Indeed, immunoblot analysis revealed a significantly reduction of total cellular H3K4m3 level after SAM depletion (Fig. 6f). However, despite a difference in total H3K4m3 level, H3K4m3 ChIP analysis of control, NAC or PQ treated-NSPCs did not reveal significant changes in terms of promoter occupancy or peak width/height/area under redox challenges (Reviewer Fig. 5A-B). Based on these findings, we concluded that

regulation of methyltransferases activity may not be the major mechanism by which SAM depletion modulates neuronal differentiation.

was shown to be most sensitive to the lowered SAM levels due to its high Km value^{7,8}. Indeed, immunoblot analysis revealed a significantly reduction of total cellular H3K4m3 level after SAM depletion (Fig. 6f). However, despite a difference in total H3K4m3 level, H3K4m3 ChIP analysis of control, NAC or PQ treated-NSPCs did not reveal significant changes in terms of promoter occupancy or peak width/height/area under redox challenges (Reviewer Fig. 5A-B). Based on these findings, we concluded that

Reviewer Figure 5. Lack of substantial changes in genome wide H3K4m3 peaks. A. Representation of H3K4m3 peak densities at transcription start sites (TSS)+/-5 kb regions from NSPCs treated with NAC (5 mM), PQ (10 μM) or control. B. Pair-wise comparisons of H3K4m3 peak height, area, or width between NSPCs treated with NAC, PQ or control. Note there is no significant deviation among differently treated samples.

As to the specificity of oxidative stress-induced IFN-I signaling on NSPC differentiation regulation, we showed that IFNβ treatment alone was sufficient to suppress neurogenic differentiation of NSPCs (Fig. 1h). Moreover, through depletion of interferon alpha and beta receptor subunit 2 (IFNAR2), a key subunit of type-I IFNAR dimer, our new experiment showed that genetic inhibition of type-I IFNAR signaling could largely restore the reduced neurogenic

differentiation capacity in PQ-treated NSPCs, supporting the direct link between IFN-1 signaling and NSPC differentiation regulation. The new results are now included as Fig. 1i and Supplementary Fig. 1d, e.

Reviewer #3 (Remarks to the Author):

Hwang et al report their findings of a molecular mechanism involving FOXO3, nuclear lamins, SAM-methyl donor and innate immunity, which they propose to form a protective response in neural stem cells upon acute oxidative stress, to maintain stemness. In more detail, the authors utilize paraquat, a ROS-generating toxin, and report activation of FOXO3 and upregulation of GNMT to deplete SAM, which they propose to disrupt the nuclear lamin maturation. Moreover, they suggest that compromised nuclear envelope integrity further activates innate-immunity through cGAS/STING-IFN-1 signaling cascade to regulate neurogenesis under oxidative stress conditions. The proposal is intriguing and the pathway component links are partially but not completely established. Despite this, the findings of GNMT-SAM-FOXO3 and lamin links to innate immunity induction are interesting.

Overall, the manuscript is well-written and experiments are mostly of good technical quality with data supporting the conclusions. However, quite a few points are essential to address:

1. The authors study cultured neural stem cells. However, whether the reported mechanistic findings are specific to NSCs or generally relevant for other cells has not been studied. Some primary non-stem cell lines should be tested, such as MEFs.

We thank the reviewer for the suggestion. We have now examined our findings in MEFs. We found that as in NSPCs, redox challenge in MEFs enhanced their expression and nuclear localization/activation of FOXO3 in response to ROS. We also documented that FOXO3 downstream effectors (GNMT, cytosolic chromatin, cGAS, IFN-I response) are all activated in ROS-treated MEFs. These new results are shown as Reviewer Fig. 6.

Reviewer figure 6. Activation of FOXO3-cGAS/STING-ISGs signaling pathway by ROS in MEFs. a. Representative WB results for indicated proteins in MEFs following 2-day treatment of H₂O₂ (500 μ M) or PQ (50 μ M). **b.** IF analysis for FOXO3 following 2-day treatment as in (a). % of cells with nuclear FOXO3 is plotted. Scale bar= 10 μ m. Mean \pm s.e.m. of 10 images. **c.** Microscopic analysis of cGAS foci and extranuclear chromatin stained for H3K4m3 following 24 h treatment of H₂O₂ (500 μ M) or PQ (50 μ M). % of cells with cGAS-GFP and H3K4m3 double positive foci is plotted. Scale bar= 10 μ m. Mean \pm s.e.m. of 10 images. **d.** qRT-PCR analysis for ISGs following 2-day treatment of H₂O₂ (500 μ M) or PQ (50 μ M). Mean \pm s.e.m. of 3 independent experiments. Statistical significance was determined by one-way ANOVA for **b-d**. **** p <0.0001, *** p <0.001, ** p <0.01, * p <0.05. Experiments of **a**, **b**, and **c** were repeated three times with similar results and representative images are shown.

2. Paraquat is the ROS-inducer and main insult used in the paper. However, it is a toxin, and acts through many mechanisms. Still no consensus exists, which are the most relevant effects in different cell types. The increase of oxygen radicals, linked to impaired iron metabolism, lipid peroxidation, and ultimately ferroptosis all contribute to cell dysfunction and death. What is the amount of cell death with the used dose of paraquat in their NSCs?

The reviewer made a couple of very important points. PQ can pose toxicity at high concentrations. But with the dosage ($\leq 10 \mu$ M) used in our experiments, it does not seem to cause significant cell death, as evidenced by our measurement of cleaved caspase 3 levels. The data is now included into Supplementary Fig. 1a. Moreover, we totally agree with the reviewer that besides the FOXO3-GNMT/SAM-lamin-cGAS/STING-IFN-I signaling cascade identified in our study, ROS may affect many other important cellular processes in NSPCs,

possibly including ferroptosis. But we also hope that the reviewer will agree with us that those endeavors lay outside the scope of the current study.

3. To use paraquat as a ROS-generator and conclude that the effects of this severe toxin are physiologically relevant is quite far fetched. In Figure 3E they use H₂O₂ for some reason, but elsewhere paraquat. H₂O₂ would be an excellent and more physiological tool for ROS effect to replicate some key findings.

The reviewers #1 and #2 also raised the same question. As detailed in the response to Reviewer #1, PQ acts through cyclic depletion of anti-oxidative buffering capacity and its effect on cellular ROS is indirect but durable (Reviewer Fig. 3)¹. By contrast, more commonly used oxidative stressor H₂O₂ is direct but rather transient in its effect. H₂O₂ might be an excellent tool of studying signaling shortly after treatment or acute physiological effect of oxidative stress. However, since NSPC differentiation in our system generally takes days, we thought that for our experimental purpose, PQ might be a better tool than H₂O₂. The usage of H₂O₂ instead of PQ was purely due to a technical consideration. This particular experiment requires co-treatment of oxidative stressor with dimedone, which is necessary for detection of sulfenylated protein under the oxidative stress condition. Since PQ treatment takes longer time to establish the redox potential, we were initially afraid that its co-treatment with dimedone might generate high cellular toxicity. But nevertheless, we have now performed new experiments by using PQ treatment. The data indicate that PQ treatment of NSPCs, as in H₂O₂ treatment, induced FOXO3 sulfenylation in a dose-dependent manner. This new piece of result is now included as Supplementary Fig. 3b.

4. Neural stem cells can be identified by IHC in vivo, in subventricular zone of mice. An in vivo paraquat-exposure and analysis of specific readouts (FOXO3 in nucleus; nuclear membrane abnormality in stem cells, IHC of IF-beta linked genes or GNMT) would point to in vivo relevance of the findings and indicate specificity / non-specificity to stem cells and strengthen the paper considerably.

This is a good suggestion. But since PQ administration and its penetration through brain barrier have not been thoroughly tested before, we instead chose tMCAO-reperfusion method, a well-accepted model of oxidative stress in the brain^{4,5}. Our analysis of NSPCs in subventricular zone regions showed increased FOXO3 nuclear localization and reduction of mature nuclear lamin following the hypoxia-reperfusion-induced oxidative challenge. In addition, immunoblot and qPCR analysis of the microdissected SVZ tissues of the challenged animals revealed the enhanced expression of FOXO3, GNMT as well as other FOXO3 downstream and ISG genes, as compared to the sham control animals. These new results are now included as Supplementary Fig. 8a-f.

5. In figure 1, the authors suggest that type I interferon response is the most enriched pathway upon PQ treatment accompanied by reduced production of TUBB3 and DCX during neuronal differentiation (Figure 1D and 1E). Figure 1D is not clear, and inadequately explained in the text and legend. Please revise. Furthermore, increased TUBB3 and DCX were detected upon NAC treatment. Do the authors observe reduced expression of IFN I signaling players upon NAC treatment with or without paraquat?

The text and legend for Fig. 1d-e have been revised to increase the clarity.

We have performed the experiments as suggested. In brief, the IFN-I gene signature was not induced by NAC, but NAC co-treatment could suppress PQ or H₂O₂-induced activation of ISGs

and TBK1 phosphorylation. The data are included as Fig. 1e-f, Supplementary Fig. 1b, 1c, and 3a.

6. A complete set of data of the gene expression profiling of differentially treated NSPCs should be added to the supplement.

The compiled gene expression profiles of control, PQ, NAC treated NSPCs are now publicly available under NCBI GEO:GSE146243. In addition, complete list of GSEA output from Fig. 1d is now included as Supplementary Table 1.

7. In Figure 1G and 1F, significant amount of IFNbeta was shown to be released from the cells upon PQ treatment and IFNbeta treatment could inhibit the production of TUBB3 during neuronal differentiation. IFNAR is the ubiquitous receptor for type I interferon cytokines. Does depletion of the receptor on NSPCs prior to PQ treatment also attenuate activation of type I interferon signalling and neuronal differentiation? This finding would further support the specificity of these defects to type I interferon cytokines. A negative control to assess for the activation of pro-inflammatory cytokines like IL-1beta or IL-6 should also be included.

This is a great suggestion and we have accordingly performed experiments to address the raised issue. First, we used CRISPR/Cas9 approach to deplete IFNAR2, a key subunit of IFNAR dimer. Our data indicated that genetic inhibition of IFNAR-signaling restored the reduced neurogenic differentiation capacity in PQ-treated NSPCs, supporting the direct link between IFN-1 signaling and NSPC differentiation regulation. These new results are now included as Fig 1i and Supplementary Fig.1d-e. In addition, we have examined the levels of IL-1b and IL-6 in response to PQ and found that both are barely expressed before and after PQ treatment (Reviewer Fig. 7).

Reviewer Figure 7. Lack of substantial increases in other pro-inflammatory cytokines. qRT-PCR analysis of PQ treated NSPCs (5 uM, 48 h) for the expression of Ifnb1, Il1b, or Il6 (n=3). *** p<0.001

8. In Figure 2, the authors show that FOXO3 depletion attenuated PQ-induced IFN I signaling. Mice deficient of FOXO family members including FOXO1, FOXO3 and FOXO4 have excessive oxidative stress which in turn decreases HSC number and their long-term regenerative potential (Tothowa et al., 2007, Cell). Do FOXO1 and FOXO4 depletion also contribute to this immune signaling restoration, or is the effect in neural stem cells FOXO3 specific?

This is a valuable point. FOXO3 is the predominant isoform expressed NSPCs as we and others have previously reported^{9,10}. Consistently, depletion of FOXO3 alone is sufficient to reverse ROS-effect in NSPCs. The other isoforms like FOXO1 and FOXO4 are expressed albeit at much lesser levels. It is fully possible that FOXO1 and/or FOXO4 could contribute or even might be the major contributor(s) of oxidative stress-induced IFN response. But we hope that the reviewer will agree with us that those are beyond the scope of current study.

9. In Figure 4, the authors observed abnormal nuclei and cGAS-GFP foci upon PQ treatment indicating compromised nuclear envelope integrity, and showed increased cGAS foci upon expression of GNMT in Figure 6K. The authors claim that nuclear DNA leakage induces

activation of cGAS/STING pathway following GNMT activation and insufficient lamin maturation. Their DNA-leakage conclusion is only based on nuclear morphology and detection of cGas/STING foci in the cells. A more direct evidence of DNA leakage following oxidative stress elevation is needed, for instance, direct immunofluorescence analysis visualizing DNA to show increased cytosolic DNA (present outside of nuclei), upon treatment with PQ or GNMT overexpression. Restoration of interferon signaling following depletion of cGAS or STING in these cells prior to oxidative stress elevation would further support the role of cGAS-STING pathway in NPSCs.

This is a good suggestion. In addition to images of DAPI stained nuclei, we have now added H3K4m3 staining images that documented chromatin leaking outside of nuclear envelope in response to PQ or upon GNMT induction. The new data are now included in Fig. 4a and 6k. In addition, through treatment of RU.521, a cGAS inhibitor¹¹, we verified the critical role of cGAS/STING pathway in ROS-induced IFN-I response. These pieces of new data are now included as Supplementary Fig. 5b-d.

10. The experiments using FOXO3-C31A mutant are excellent, providing clear views to the mechanisms. Could the C31A mutant of FOXO3 induce IFN-beta upon PQ?

This is a valid point. In our new experiment, we found that PQ-induced IFN-beta expression in FOXO3-C31A mutant transduced NSPCs was reduced as compared to FOXO3-WT transduced control cells. This result was added to Fig 3i.

11. The finding of the link of low SAM, activation of cGAS/sting and induction of IFbeta is solid and interesting. Whether all their findings are mechanistically linked, or whether several pathways may lead to IFNbeta induction, is not clear. However, paraquat is a toxin, with several deleterious effects. Whether the pathways reported actually are a physiological mechanism that regulates neurogenic potential is not verified by the paper. The discussion of relevance to aging and regulation of NSC neurogenesis, without any in vivo data, is too far fetched and should be omitted.

The reviewer raised important points. In the current study, PQ and H₂O₂ were used as chemical agents to induce oxidative stress. However, it is worthy to note that the ROS-FOXO3-GNMT/SAM-lamin-IFN-I cascade illustrated in our study is not PQ or H₂O₂-specific. We showed that ROS-mediated IFNbeta induction can be recapitulated by enforced expression of an active mutant FOXO3 (FOXO3-TA) (Fig. 2e-h), GNMT (Fig. 6f-n), or depletion of ICMT (Fig. 5g-i). We further demonstrated that depletion of either FOXO3 or GNMT reversed the ROS effect and was sufficient to restore their neurogenic potential in the PQ-treated NSPCs (Fig. 7g-j), supporting that the cascade is mechanistically linked. In addition, our new data through analyzing a well-characterized tMCAO model revealed that acute oxidative stress induced a time-dependent increase of nuclear FOXO3 and loss of 8D1-positive mature lamin expression in NESTIN+ progenitors (Supplementary Fig.8c, d), suggesting that our findings are physiologically relevant. But we totally agree with the reviewer that extensive in vivo studies will be needed in the future to verify the significance of our findings in the context of physiological/pathological aging and related neurogenesis. We have modified our text accordingly.

12. A quantification of western blotting results and kDa molecular mass for proteins identified should be provided. As many westerns (e.g. 2b) are of suboptimal quality, the whole gels should be included in supplementary data.

We have included uncropped gels under source data file.

Reviewer #4 (Remarks to the Author):

The manuscript by Hwang et al. proposes a role for FOXO3-mediated regulation of lamin post-translational modification (carboxymethylation) on the response to cellular stress that activates the cGAS-STING pathway and the downstream interferon (IFN) response. In addition, they propose that activation of this cascade suppresses neurogenic differentiation of neural stem/progenitor cells (NSPCs). The study touches a subject that is understudied and that has received high attention in recent years: the connection between cellular stress, the IFN response, and stem cell differentiation potential. Overall, the study is significant for the field, describing new molecular mechanisms linking oxidative stress to activation of inflammatory cascades such as the IFN response. It also highlights the importance of proper post-translational processing of lamins not only for nuclear function but also for processes that take place at the cytoplasm, such as cGAS/STING/IFN pathway activation, and that are key for cellular homeostasis. The findings are innovative and of potential high impact across different fields.

We thank the reviewer for the thoughtful summary and positive comments.

Strengths:

- The manuscript is well-written, the experimental design is for the most part appropriate to test the different hypotheses, and the conclusions are in accordance with the results. In addition, the proper controls are included, with some exceptions noted below.
- Strong evidence is provided for an impact of oxidative stress inhibiting NSPCs' differentiation, as well as RNAseq evidence for activation of an IFN response.
- Convincing data also support the notion that translocation of FOXO3 into the nucleus and activation of downstream effectors mediates IFN response activation, with Cys31 sulfenylation and Thr32 phosphorylation being important.
- Moreover, the authors make a clear-cut connection with lamins post-translational processing of their C-terminal -CAAX motif. In particular, the C-terminus of lamins seems to play a role in the activation of IFN response by PQ. Proper processing of lamins is critical to maintain IFN pathway under control, and hindering processing activates the pathway. A special role is shown for carboxymethylation of terminal Cys in lamins, which is inhibited upon PQ due to reduction in SAM levels triggered by upregulation of GNMT. These findings lead the authors to conclude that GNMT upregulation and SAM depletion elicit deficiencies in lamins carboxymethylation that cause nuclear fragility and activation of cGAS/STING and downstream IFN signaling. The importance of lamins in this process is clearly demonstrated by expression of a mutant lamin A protein that lacks the C-terminus. If lamins cannot be methylated, the problem of overexpression of GNMT activated the IFN response is solved.
- The authors elegantly close the circle by showing a connection between FOXO3 activation and GNMT induction under oxidative stress conditions, which plays a role triggering the IFN response and downstream effectors. In addition, they demonstrate that FOXO3 and GNMT upregulation upon PQ treatment hinder the differentiation of NSPCs.

Weaknesses:

1. Experiments of oxidative stress were performed only with paraquat. Other ways to induce oxidative stress could be tested to make sure that this is not a paraquat-specific effect. Along the same lines, the RNAseq on NAC treated samples is missing. Showing that NAC does not activate the IFN response would reinforce the idea that the trigger of this response is oxidative stress. In the RNAseq analysis, did they find upregulation of FOXO targets upon PQ treatment?

We appreciate this point and the suggestions. During our revision, we have performed new experiments to confirm our key findings. Particularly, we have conducted the in vivo experiment

using transient middle cerebral artery occlusion (tMCAO)-reperfusion, is a well-characterized brain oxidative stress model^{4,5}. Our analysis of NSPCs at SVZ regions revealed a time-dependent increase of nuclear FOXO3 localization following stress challenge that was accompanied by a loss of 8D1-positive mature lamin expression. Immunoblot and qPCR analysis of the microdissected SVZ tissues of the challenged animals further indicated the enhanced expression of FOXO3, GNMT as well as other FOXO3 downstream and ISG genes. These new pieces of results are now included as Supplementary Fig 8a-f. To fully rule out the possibility that our findings are a PQ-specific effect, we have also performed anti-oxidant NAC rescue experiments as suggested by the reviewer. The new results are now included in Fig. 1f and Supplementary Fig. 1b, 1c, and 3a.

In addition, we have now included the RNAseq results for NAC treated NSPCs. Under NAC treatment IFN-I response genes were not induced (Fig. 1e). Also, GSEA analysis for FOXO target genes showed PQ-treatment induced FOXO3 and its targets. These analysis results are included as Supplementary Fig. 2a, b.

2. Depletion of FOXO3 reduces the IFN response in response to PQ but also basal IFN response, suggesting that FOXO3 might regulate the IFN response in all contexts, independent of oxidative stress. This is in fact the case when the authors express the active form of FOXO3. Thus, the conclusion by the authors that “FOXO3 is directly responsible for oxidation stress-induced IFN-I activation” is not totally accurate. FOXO3 seems to activate the IFN response independent of oxidative stress. A question that arises from these data is whether FOXO3 would be a key regulator of the IFN response due to DNA damage, mitochondrial dysfunction, or other stresses. Additional experimental data that addresses whether this is a general mechanism or specific for oxidative stress would improve the quality/significance of the manuscript.

This is an excellent point and we thank the reviewer for the suggestion. DNA damage or mitochondrial dysfunction are known stressful causes of IFN-I response. We have now examined whether FOXO3 depletion could also suppresses IFN-I response in those stress conditions. In brief, our analysis of DNA damage (zeocin treatment) or energy stress condition upon mitochondria ATP synthase inhibition (oligomycin A) showed that depletion of FOXO3 could also attenuate IFN-I response induced by those stress inducers. These new results are included as Supplementary Fig. 2c-e.

3. The cGAS/STING/IFN pathway is considered upstream of the IFN response, however, while the IFN response is monitored extensively, less is done to test the activation of cGAS/STING pathway. To make the claim that cGAS/STING pathway is upstream of all the phenotypes, inhibition of this pathway should be performed.

We agree with the reviewer and performed new experiments through pharmacological approach of pathway inhibition. By treatment of RU.521, a cGAS inhibitor¹¹ we further confirmed that cGAS/STING pathway plays a critical role in mediating oxidative stress-induced IFN response. These results are now included as Supplementary Fig. 5b-d.

Other comments:

1. Why do they use H₂O₂ only in experiment in Fig 3e? Is there FOXO3 sulfenylation upon PQ treatment?

The three other reviewers also raised the similar concern. In brief, the usage of H₂O₂ instead of PQ there was purely due to a technical consideration. PQ acts through cyclic depletion of anti-oxidative buffering capacity and its effect on cellular ROS is indirect but durable (Reviewer Fig.

3) ¹. By contrast, more commonly used oxidative stressor H₂O₂ is direct but rather transient in its effect. This particular experiment requires co-treatment of oxidative stressor with dimedone, which is necessary for detection of sulfenylated protein under the oxidative stress condition. Since PQ treatment takes much longer time to achieve the redox potential, we were afraid that its co-treatment with dimedone might generate high cellular toxicity. But nevertheless, we have now performed new experiments by using PQ treatment. The data indicate that PQ treatment of NSPCs, as in H₂O₂ treatment, induced FOXO3 sulfenylation in a dose-dependent manner. This new piece of result is now included as Supplementary Fig. 3b.

2. In Figure 3h and 3i, the control of PQ treatment in FOXO3 KO cells is missing.

Sorry for the error. We have added the controls into Figure 3h-j.

3. In Figures 7g-k it is not clear what High and Low means. Are they different concentrations of PQ?

Thanks for pointing it out. We have re-labelled the figure. It was meant to indicate high redox potential (high oxidative stress) or low redox potential (low oxidative stress) conditions.

Literature Cited

1. Wiemer, M. & Osiewacz, H.D. Effect of paraquat-induced oxidative stress on gene expression and aging of the filamentous ascomycete. *Microb Cell* **1**, 225-240 (2014).
2. Essaghir, A., Dif, N., Marbehant, C.Y., Coffey, P.J. & Demoulin, J.B. The transcription of FOXO genes is stimulated by FOXO3 and repressed by growth factors. *J Biol Chem* **284**, 10334-10342 (2009).
3. Hwang, I. *et al.* FOXO protects against age-progressive axonal degeneration. *Aging Cell* **17** (2018).
4. Murakami, K. *et al.* Mitochondrial susceptibility to oxidative stress exacerbates cerebral infarction that follows permanent focal cerebral ischemia in mutant mice with manganese superoxide dismutase deficiency. *J Neurosci* **18**, 205-213 (1998).
5. Peters, O. *et al.* Increased formation of reactive oxygen species after permanent and reversible middle cerebral artery occlusion in the rat. *J Cereb Blood Flow Metab* **18**, 196-205 (1998).
6. Garrett-Bakelman, F.E. *et al.* Enhanced reduced representation bisulfite sequencing for assessment of DNA methylation at base pair resolution. *J Vis Exp*, e52246 (2015).
7. Mentch, S.J. & Locasale, J.W. One-carbon metabolism and epigenetics: understanding the specificity. *Ann N Y Acad Sci* **1363**, 91-98 (2016).
8. Krivtsov, A.V. & Armstrong, S.A. MLL translocations, histone modifications and leukaemia stem-cell development. *Nat Rev Cancer* **7**, 823-833 (2007).
9. Paik, J.H. *et al.* FoxOs cooperatively regulate diverse pathways governing neural stem cell homeostasis. *Cell Stem Cell* **5**, 540-553 (2009).
10. Renault, V.M. *et al.* FoxO3 regulates neural stem cell homeostasis. *Cell Stem Cell* **5**, 527-539 (2009).
11. Vincent, J. *et al.* Small molecule inhibition of cGAS reduces interferon expression in primary macrophages from autoimmune mice. *Nat Commun* **8**, 750 (2017).

REVIEWERS' COMMENTS

Reviewer #1 (Remarks to the Author):

The authors have addressed my concerns. Just one minor point that should be addressed: the heatmap in Fig 1e still needs a color key indicating what the red/white/blue values represent.

Reviewer #2 (Remarks to the Author):

All of my concerns have been addressed.

Reviewer #3 (Remarks to the Author):

I appreciate it that the authors have made an effort to respond satisfactorily to most of our major comments. I do not have further concerns or criticism.

Reviewer #4 (Remarks to the Author):

In the new version of the manuscript, the authors have addressed the questions raised, both by including new experiments and improving discussion of the results. The quality of the manuscript has significantly improved. The findings are likely to have an impact across different fields.

Point by point response to REVIEWERS' COMMENTS

Reviewer #1 (Remarks to the Author):

The authors have addressed my concerns. Just one minor point that should be addressed: the heatmap in Fig 1e still needs a color key indicating what the red/white/blue values represent.

In Figure 1e, the heatmap colors represent z-score [a.k.a. standard score, $z=(\text{value}-\text{mean})/\text{standard deviation}$]. A color scale bar is included underneath. In the color key, the red represents a positive value (upregulated), white being no change, and the blue represents a negative value (downregulated). We have now included the description into the figure legend.

Reviewer #2 (Remarks to the Author):

All of my concerns have been addressed.

Reviewer #3 (Remarks to the Author):

I appreciate it that the authors have made an effort to respond satisfactorily to most of our major comments. I do not have further concerns or criticism.

Reviewer #4 (Remarks to the Author):

In the new version of the manuscript, the authors have addressed the questions raised, both by including new experiments and improving discussion of the results. The quality of the manuscript has significantly improved. The findings are likely to have an impact across different fields.